# Metabolic specialization drives reduced pathogenicity in *Pseudomonas aeruginosa* isolates from cystic fibrosis patients

**Bjarke Haldrup Pedersen**[1], **Filipa Bica Simões**[2], **Ivan Pogrebnyakov**[1], **Martin Welch**[3], **Helle Krogh Johansen**[2,4], **Søren Molin**[1], **Ruggero La Rosa**[1,2]*

1 The Novo Nordisk Foundation Center for Biosustainability, Technical University of Denmark, Kongens Lyngby, Denmark, 2 Department of Clinical Microbiology 9301, Rigshospitalet, Copenhagen, Denmark, 3 Department of Biochemistry, University of Cambridge, Cambridge, United Kingdom, 4 Department of Clinical Medicine, Faculty of Health and Medical Sciences, University of Copenhagen, Copenhagen, Denmark

* rugros@biosustain.dtu.dk

**Data Availability Statement:** The authors declare that all data necessary for supporting the findings of this study are enclosed in this paper (S1–S4

## Abstract

Metabolism provides the foundation for all cellular functions. During persistent infections, in adapted pathogenic bacteria metabolism functions radically differently compared with more naïve strains. Whether this is simply a necessary accommodation to the persistence phenotype or if metabolism plays a direct role in achieving persistence in the host is still unclear. Here, we characterize a convergent shift in metabolic function(s) linked with the persistence phenotype during *Pseudomonas aeruginosa* colonization in the airways of people with cystic fibrosis. We show that clinically relevant mutations in the key metabolic enzyme, pyruvate dehydrogenase, lead to a host-specialized metabolism together with a lower virulence and immune response recruitment. These changes in infection phenotype are mediated by impaired type III secretion system activity and by secretion of the antioxidant metabolite, pyruvate, respectively. Our results show how metabolic adaptations directly impinge on persistence and pathogenicity in this organism.

## Introduction

Difficult to treat bacterial infections are increasing around the world [1]. While antibiotic resistance is a major cause of treatment failure, other less characterized mechanisms rooted in the complexity of the host–pathogen interactions are also substantial contributors to persistence [2]. Genetic variants with high tolerance to stresses, host and immune evasion capabilities, and low virulence are often specifically selected for in persistent infections since they provide higher within-host fitness [3,4]. In people with cystic fibrosis (pwCF), for example, opportunistic pathogens such as *Pseudomonas aeruginosa* colonize the airways and establish persistent infections that can last for more than 30 years. During the infection, the bacteria differentiate into heterogeneous populations specifically evolved for the host microenvironment [5–8]. Surprisingly, antibiotic resistance is not the initial driver of persistence, since bacteria retain antibiotic susceptibility for years after colonization [9]. Seemingly, these populations

Data). All genomic data is publicly available through the SRA database and has been published previously by Marvig et al., (2015), doi:10.1038/ng. 3148.

**Funding:** The work at the Novo Nordisk Foundation Center for Biosustainability is supported by the Novo Nordisk Foundation www. novonordiskfonden.dk (grant number NNF20CC0035580). This work was supported by the UK Cystic Fibrosis Trust www.cysticfibrosis. org.uk (grant number SRC017 - MW, SM, HKJ) and the Independent Research Fund Denmark/ Natural Sciences www.dff.dk (grant number 9040-00106B - SM). HKL was supported by the Novo Nordisk Foundation www.novonordiskfonden.dk (Challenge grant NNF19OC0056411). The funders had no role in study design, data collection and analysis, decision to publish, or preparation of the manuscript.

**Competing interests:** The authors have declared that no competing interests exist.

**Abbreviations:** ALI, air–liquid interface; COG, Clusters of Orthologous Groups; HCA, hierarchical cluster analysis; HCN, hydrogen cyanide; LPS, lipopolysaccharide; PCA, principal component analysis; PDHc, pyruvate dehydrogenase complex; PFA, paraformaldehyde; PQS, Pseudomonas quinolone signal; pwCF, people with cystic fibrosis; ROS, reactive oxygen species; SCFM2, synthetic cystic fibrosis medium 2; T3SS, type III secretion system; TEER, transepithelial electrical resistance.

use mechanisms of persistence such as biofilm formation, loss of flagella and virulence factors, growth rate reduction, immune escape, and metabolic specialization to "hide" from the immune system and withstand the antibiotic treatment [2]. These mechanisms usually fall below the detection radar since, in simple laboratory conditions, without the intrinsic complexity of the host environment and its interactions with the bacteria, no assumptions on the persistence of such populations can be made. However, these mechanisms are relevant for the overall persistence of the infecting population, possibly being the main drivers of the initial host colonization before the insurgence of antibiotic resistance.

One hypothesis is that metabolic specialization strongly influences the host–pathogen interactions and leads to persistence [10,11]. In many infections, the nutrient composition of the host microenvironment provides an environmental cue for bacterial pathogens to activate their virulence repertoire [11]. Abnormalities in calcium homeostasis, nutrient limitation, or change in pH can, for example, trigger the type III secretion system (T3SS) cascade in several bacterial species, activating the secretion of virulence effectors promoting colonization of the host [12–15]. In *P. aeruginosa*, the T3SS has a key role during infection, since through its injectosome and secreted factors, it subverts the host cell machinery influencing both invasion, growth, and host immune response [12,16,17]. Similarly, metabolism-dependent processes such as biofilm production and mucoidy provide shielding from antibiotics and immune cells and are generally associated with worse prognoses [18–20]. Furthermore, crosstalk between bacteria and immune cells through exchange of bacterial and host metabolites has additionally revealed the importance of metabolism in determining the outcome of an infection [21–23]. Importantly, metabolism is not a static function; it dynamically changes to support cell growth and virulence specifically for each host and type of infection. Clinical strains of *P. aeruginosa* that infect pwCF have been shown to modify their metabolic preference to accommodate the specific nutrient composition of the airways [24,25]. Auxotrophy, specialized assimilation of carbon sources, secretion of high value metabolites and differential oxygen requirements of adapted clinical strains ensure appropriate functionality of the cell and support the phenotype requirements in the host [10]. However, it is still unclear if and how metabolic specialization directly contribute(s) to persistence [24,26]. Furthermore, the specific selective forces (for example, antibiotic treatment or the immune system) leading to metabolic specialization still remain uncharacterized. A few examples of laboratory studies in *P. aeruginosa* and *Escherichia coli* have suggested the involvement of specific metabolic mutations with changes in antibiotic susceptibility and virulence [27–31]. However, limited knowledge is available on their relevance in clinical isolates of *P. aeruginosa* during an infection. Moreover, the extent to which metabolic specialization per se provides a specific fitness advantage or if it is merely a downstream effect of accommodating other essential phenotypes still remains unknown. For example, it has been shown that overexpression of multidrug efflux pumps causing antibiotic resistance, can lead to rewired metabolism, thereby compensating for the associated fitness cost [32]. However, if such a mechanism is generalizable and whether it specifically contributes to persistence remains still unexplored. Importantly, previous metabolic characterizations of clinical strains were carried out on only a limited number of isolates and/or on bacterial cultures at one specific growth phase (exponential or stationary phase) lacking the dynamics of metabolic processes [24,33,34]. Because of technical challenges related to the complexity of dynamic metabolomic analysis and its interpretation, large-scale analyses of populations of clinical isolates that account for the inherent dynamics of metabolism have so far been lacking. Moreover, the effect of metabolic specialization on the host–pathogen interactions remains unclear, limiting the understanding of its contribution to persistence [2,35]. It is, therefore, crucial to systematically identify and characterize new mechanisms of metabolic specialization to build a comprehensive understanding of the contribution of metabolism to persistence.

Such efforts will provide new understanding of treatment failure and unravel new pathogen vulnerabilities and therapeutic options, which are currently overshadowed by the focus on increasing antibiotic resistance.

Here, we identify and characterize molecular mechanisms of metabolic specialization occurring in clinical strains of *P. aeruginosa* associated with persistence in pwCF. We further show that these mechanisms have an impact on the relationship between the host and the pathogen. By analyzing, in detail, the metabolic and proteomic profiles of clinical strains from pwCF at different stages of within-host evolution, we identified distinct metabolic configurations characterized by CF-specific nutrient assimilation and secretion patterns. These changes in metabolism and proteome allocation are directly linked with mutations affecting key metabolic genes. We also characterize 1 specific mechanism of metabolic specialization involving the pyruvate dehydrogenase complex (PDHc), which is essential for the processive flux of pyruvate through central carbon metabolism. Surprisingly, recombinant strains containing single mutations in the PDHc show decreased infection capabilities and are associated with inflammation in an air–liquid interface (ALI) infection model system. These strains display a chronic infection phenotype and an increased secretion of pyruvate, which is an important scavenger of reactive oxygen species (ROS) and has known anti-inflammatory properties [36–38]. Importantly, we show that these mutations are widespread in clinical isolates of *P. aeruginosa* from different patients and infection scenarios. This suggests that metabolic specialization might be specifically selected for during the early stages of an infection to limit host-dependent inflammation. Altogether, these results provide a rationale for metabolic specialization during CF airway infections.

## Results

### Strain collection for metabolomic analyses

To identify molecular mechanisms of metabolic specialization which modify the interactions between the host and the pathogen occurring during the infection of pwCF, we selected from our longitudinal collection of 474 *P. aeruginosa* clinical strains [39] pairs of early (*_E; first *P. aeruginosa* isolated from the person) and late (*_L) clinical isolates, longitudinally isolated from eight different pwCF (Fig 1A). Previous studies have shown that slow growth is an important phenotype associated with long term adaptive evolution to the CF airways, appearing within the first 2 to 3 years of the infection [9]. Therefore, we used growth rate as a proxy for within-host evolution. This allowed us to increase the odds of selecting strains presenting unexplored mechanisms of metabolic specialization. Accordingly, we investigated metabolic adaptation in a diverse collection of 16 clinical strains from 8 distinct pwCF (Fig 1A).

As by selection criteria, when grown in synthetic cystic fibrosis medium 2 (SCFM2) (which has been formulated to resemble the CF mucus composition [40]), early isolates showed a high growth rate, comparable with the reference strain, PAO1. Conversely, late isolates showed a 2.2- to 5.1-fold reduction in growth rate, in addition to a lower maximal optical density (paired *t* test, *p* = 0.002) (Fig 1A and 1B). Moreover, some late isolates were unable to grow in minimal medium in presence of single carbon sources such as glucose, lactate, or succinate, indicative of a metabolic constraint such as auxotrophy [41] (Fig 1C).

The genome of each isolate had been previously sequenced, and their genotype and phylogenetic relationship determined [39]. Specifically, each pair of early and late clinical isolates are phylogenetically related, share a common ancestor, and belong to the same "clone type" (defined as genomes sharing more than 10,000 SNPs) (Fig 1A). Each pwCF was colonized by a distinct clone type with evolutionary histories spanning between 1.4 and 7 years (S1 Table). On average, the difference in mutations between early and late strains was 46 ± 5 with the

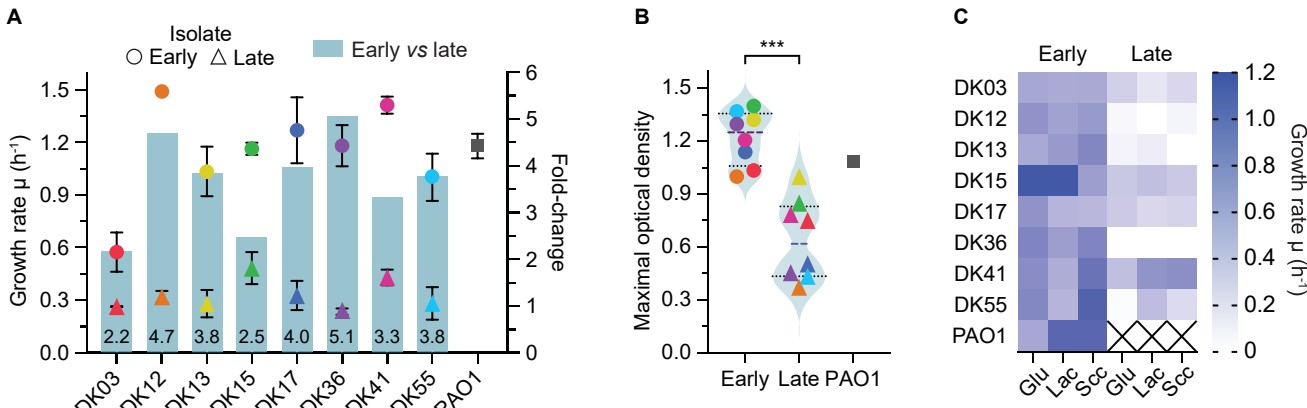

**Fig 1. Collection of *P. aeruginosa* clinical strains.** (A) Growth rate (hour$^{-1}$) of early (circles), late (triangles), and PAO1 (square) strains colored by their clone type. The fold change of growth rate from early to late is shown as bar chart on the right y axis and as number value at the base of each bar. (B) The maximal optical density (maxOD) of all isolates, grouped by early vs. late comparison. Statistical significance was calculated by unpaired Welch *t* test and indicated as *** ($p < 0.001$). (C) Growth rate (hour$^{-1}$) of each strain in M9 minimal media containing 20 mM of a single carbon source glucose (Glu), lactate (Lac), or succinate (Scc). The data underlying this figure can be found in S4 Data. maxOD, maximal optical density.

exception of strains of the DK36 clone type which show an increased number of mutations (402 mutation differences) due to hypermutation (S1 Table) [39]. As previously reported [42], each single isolate represents the most abundant representative of the population from a sputum sample. These selection criteria, therefore, allowed us to analyze isolates from different individuals with distinct clinical and evolutionary histories, rather than focusing on the heterogeneous population present in pwCF.

## Within-host evolution selects for specialized metabotypes

To evaluate the degree of metabolic specialization in the clinical isolates, dynamic exo-metabolomics was performed on cells growing in SCFM2 (S1 Data). This approach allowed for analysis of the assimilation and secretion patterns of specific metabolites during the growth of each isolate, providing a dynamic profile of their metabolic activity. Principal component analysis (PCA), k-means clustering, and hierarchical cluster analysis (HCA) of the extracellular metabolomes revealed the presence of 3 distinct specialized metabolic configurations in late strains, hereafter defined as "adapted metabotypes" (DK15 and DK55 metabotype 1; DK12 metabotype 2; DK03, DK13, and DK36 metabotype 3). These metabotypes separate the late isolates from each respective early isolate and from those strains not exhibiting metabolic changes, hereafter referred to as "naïve metabotypes" (Fig 2A). Note that whereas PCA emphasizes the largest differences in the metabolomic profiles, the HCA clearly separates the early strains from the late strains, indicating a certain degree of metabolic specialization in all late isolates (Fig 2B). Indeed, both the hierarchy of assimilation (metabolite half-lives, OD$_{50}$) which represents the order of assimilation of the available carbon sources, and the secretion of metabolites differ between early and late isolates (Fig 2C and 2D). Of note, late strains of DK12, DK36, and DK55 secreted high amounts of pyruvate which is a key metabolite connecting glucose metabolism to the TCA cycle. Detailed information on the specific metabolic preferences of each metabotype is presented in S1 Text. Furthermore, the net balance between assimilated and secreted metabolites (total mM) varies between early and late strains (Fig 2E), which positively correlates with the lower biomass of the late isolates (Pearson's $r = 0.9$; $p = 0.0023$) (Figs 1B and 2E). In other words, a high percentage of assimilated carbon sources are catabolized in the late isolates and secreted back into the culture medium, thus limiting their availability for

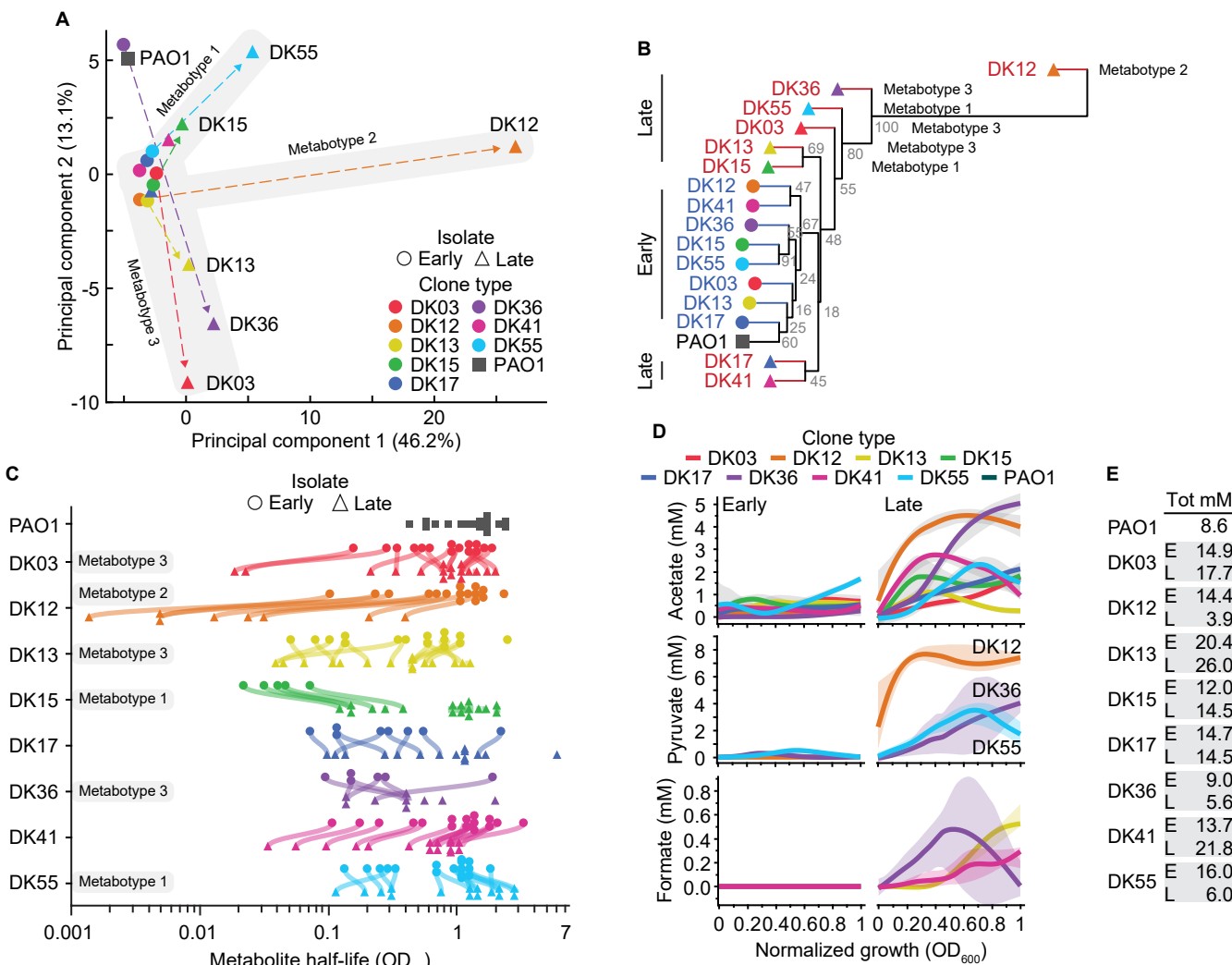

**Fig 2. Exo-metabolomics analysis reveals adapted metabotypes.** (A) PCA showing separation of strains based on their time-resolved exo-metabolomes. Strains are shown as circles (early), triangles (late), and square (PAO1), with dashed arrows indicating the most notable trajectories from early to late, colored by clone type. Metabotypes were designated based on iterative k-means clustering analysis and HCA. See Material and methods for details. (B) HCA of the exo-metabolomics data, showing a general separation between early and late strains. Branches for early isolates are indicated in blue, late isolates in red and PAO1 in black. Accuracy of the HCA was tested by bootstrapping where gray values within the branches represent the % of bootstrap values for 10,000 replicates. (C) Assimilation hierarchies of the analyzed metabolites. Each symbol represents the half-life ($OD_{50}$) of a specific metabolite while each line connects the same metabolite for each pair of early and late isolates. Missing connecting curves indicates that either the strain did not assimilate the metabolite, or that the $OD_{50}$ was outside the analyzed assimilation window. (D) Secretion plots showing variations in the concentration of acetate, pyruvate, and formate (mM) relative to normalized growth ($OD_{600}$). Shaded areas indicate the 95% confidence intervals. (E) Table showing the total amount (mM) of carbon sources assimilated for each strain. Secreted metabolites were subtracted from the total to account for their excretion in the medium. Clone types are separated by gray boxes with "E" indicating early and "L" indicating late isolate. The data underlying this figure can be found in S1 and S4 Data. HCA, hierarchical cluster analysis; PCA, principal component analysis.

biomass production. This suggests either a specific metabolic configuration for the late isolates —the objective of which is not biomass accumulation—or an apparent inefficient metabolism.

## Changes in proteome allocation supports the metabolic specialization of clinical isolates

To test the hypothesis that the observed metabolic specialization is rooted in changes in expression of proteins involved in cellular metabolism, we analyzed the proteome of each clinical strain. To this end, we performed whole cell proteomics to compare protein expression

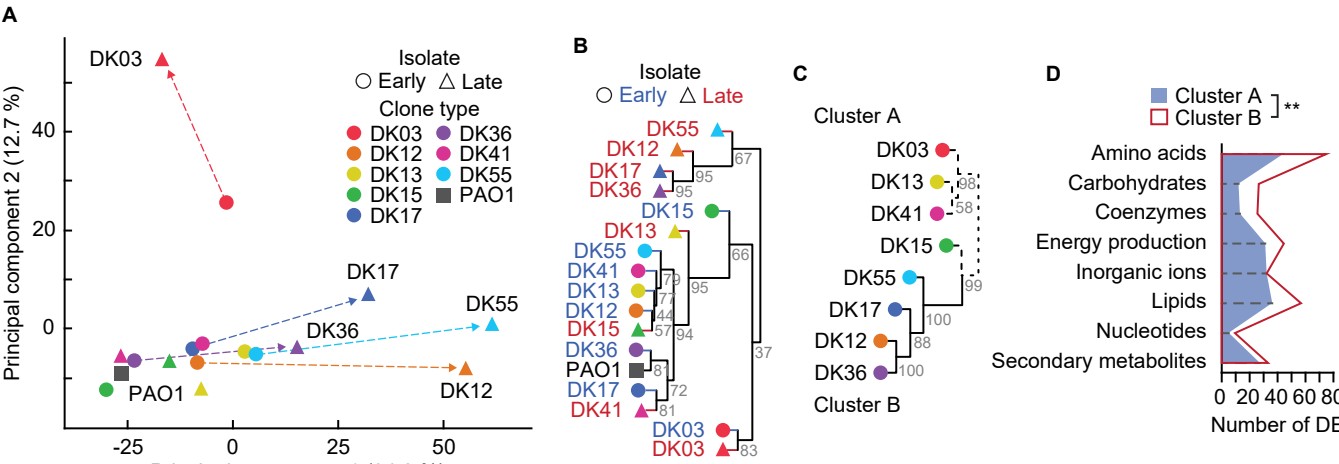

**Fig 3. Proteome composition changes support metabolism specialization.** (A) PCA showing separation of strains based on their mid-exponential proteomes. Strains are shown as circles (early), triangles (late), and square (PAO1), with dashed arrows indicating most notable trajectories from early to late, colored by clone type. (B) HCA of the proteomic data containing 2,061 proteins identified, showing a general separation between early and late strains. Branches for early isolates are indicated in blue, late isolates in red, and PAO1 in black. Accuracy of the HCA was tested by bootstrapping where gray values within the branches represent the % of values for 10,000 replicates. (C) HCA of the 235 differentially expressed proteins when comparing late vs. early strains within the COG categories related to metabolism. Proteomes are represented by dashed lines for "cluster A" and continuous lines for "cluster B." Accuracy of the HCA was tested by bootstrapping where gray values within the branches represent the % of bootstrap values for 10,000 replicates. (D) Parallel plots of the number of DE proteins within the COG categories related to metabolism in cluster A (blue, shaded) and B (red, transparent). For each category, the mean of each cluster is indicated. The difference in the total number of DE proteins between clusters when matching by category was computed by two-tailed paired *t* test where ** indicates *p* = 0.0047. The data underlying this figure can be found in S2 Data. COG, Clusters of Orthologous Groups; DE, differentially expressed; HCA, hierarchical cluster analysis; PCA, principal component analysis.

profiles between PAO1, early, and late strains (comparisons being early versus PAO1, late versus PAO1, and late versus early). This allowed us to evaluate proteome changes in the clinical isolates relative to a laboratory reference strain (early versus PAO1, late versus PAO1) and to identify in the clinical isolates possible molecular mechanism(s) underpinning the metabolic specialization we observed (late versus early). Of the 2,061 proteins identified, 740 were differentially expressed in at least 1 comparison (early versus PAO1 or late versus early) (S2 Data). Similar to the metabolomic analysis, when comparing the normalized expression profile of all identified proteins, PCA readily separated the proteome of the late isolates from that of the early isolates, suggesting that changes in metabolite profiles might be rooted in the proteome (Fig 3A and 3B). Most of the early and some of the late proteomes co-localized with the reference strain PAO1, indicating little or no changes in proteome allocation (Fig 3A and 3B). However, the proteome of the late isolates of lineages DK12, DK17, DK36, and DK55 separated from the respective early strains to form a cluster of adapted proteomes (Fig 3A and 3B). Notably, the DK03 strains formed an independent cluster, indicative of a lineage-specific proteomic signature (Figs 1A, 3A, and 3B).

A similar relationship between the proteomes is obtained through HCA, specifically when considering the 235 differentially expressed proteins (late versus early comparison) belonging to the Clusters of Orthologous Groups (COG) categories involved in metabolism (Fig 3C). When comparing differentially expressed proteins between late versus early strains, 2 clusters of proteomes are identified. Cluster A represents proteomic changes in clone types without any apparent metabolic specialization. In contrast, cluster B represents proteomic changes in strains which underwent metabolic specialization in the adaptive evolution process from early to late (DK12, DK36, and DK55). This result supports our hypothesis that metabolic specialization is likely rooted in changes in the expression of proteins involved in cellular metabolism (Fig 3C). Notably, cluster B is characterized by a significantly higher number of differentially

expressed proteins in the COG categories involved in metabolism relative to cluster A (two-tailed paired *t* test *p* = 0.0047) (Figs 3D and S1A). Similarly, several categories of proteins involved in the metabolism of amino acids, fatty acids, and sugars are statistically enriched in early and late strains (S1B Fig) providing a molecular explanation for the altered hierarchy of assimilation of the carbon sources and the reduced growth rate of the adapted metabotypes (Fig 2C and 2D). Detailed information on the convergent expression at the pathway level of metabolic proteins is presented in S1 Text (S2 Fig).

Although late strains of DK12, DK36, and DK55 belong to different metabotypes, their proteomes move in the same direction (Fig 3A–3C). This indicates that largely similar proteomes can sustain distinct metabolic configurations which ultimately depend on the metabolic fluxes thorough specific pathways and on their regulation. We note that, in the case of the DK17 strains comparison (cluster B) (Fig 3A–3C), although our metabolomic analysis did not detect any major metabolic rewiring (Fig 2A and 2B), proteome reorganization could be related to other metabolic processes which were not analyzed in our study.

## Changes in virulence traits during within patient evolution

To evaluate the relationship between metabolic specialization and virulence, we analyzed which differentially expressed proteins were enriched when comparing late versus early isolates. Interestingly, most of the changes (based on their KEGG and GO categories) were related to adaptation to the infection environment, redox balance, and virulence (Figs 4A and S1B). For example, most of the lineages showed expression changes in phenazine biosynthesis (KEGG) and secondary metabolite(s) biosynthetic process (GO), both of which are deeply involved in redox-balance, cell homeostasis, metabolism, and virulence [43,44].

Several changes in protein expression are already apparent in early strains relative to PAO1 indicating a different response of clinical strains to the airway-like conditions accompanying growth in SCFM2 (Fig 4A). Specifically, early strains show increased expression of proteins involved in alginate production, hydrogen cyanide (HCN), phenazine biosynthesis, and PQS (the *Pseudomonas* quinolone signal), and decreased expression of proteins involved in flagella biogenesis, lipopolysaccharide (LPS) O-antigen metabolism, and pyoverdine production, which are all hallmarks of an acute infection phenotype (Fig 4B). Early stage colonization and acute infection are thought to require expression of several virulence factors promoting host tissue injury and immune response impairment [45]. By contrast, when comparing late versus early strains, the adapted metabotypes—and specifically strains belonging to cluster B (DK12, DK17, DK36, and DK55)—show convergent up-regulation of proteins related to alginate, phenazine, PQS, and the type VI secretion system (T6SS), suggesting that metabolism plays an active role in regulating virulence and the chronic infection phenotype (Figs 4C and S4). Reduced virulence and cytotoxicity are advantageous for long-term infection and the establishment of a persistent infection [46]. Interestingly, the DK12 late strain also shows lower expression (-3-fold) of the T3SS toxin, ExoT, which is known to play a role in preventing phagocytosis, in the induction of cytoskeletal reorganization, and in host cell apoptosis [16]. Overall, the identified pattern of proteomic changes suggests that early isolates show reduced expression of virulence factors relative to PAO1, with an even greater reduction being associated with the late isolates. This is consistent with their persistence during CF airway infections [46].

## *aceE* and *aceF* mutations leads to metabolic specialization and impaired virulence

From the previously published genome sequences [39], we searched for mutations in the genome of each clinical isolate which might explain the secretion of pyruvate and the

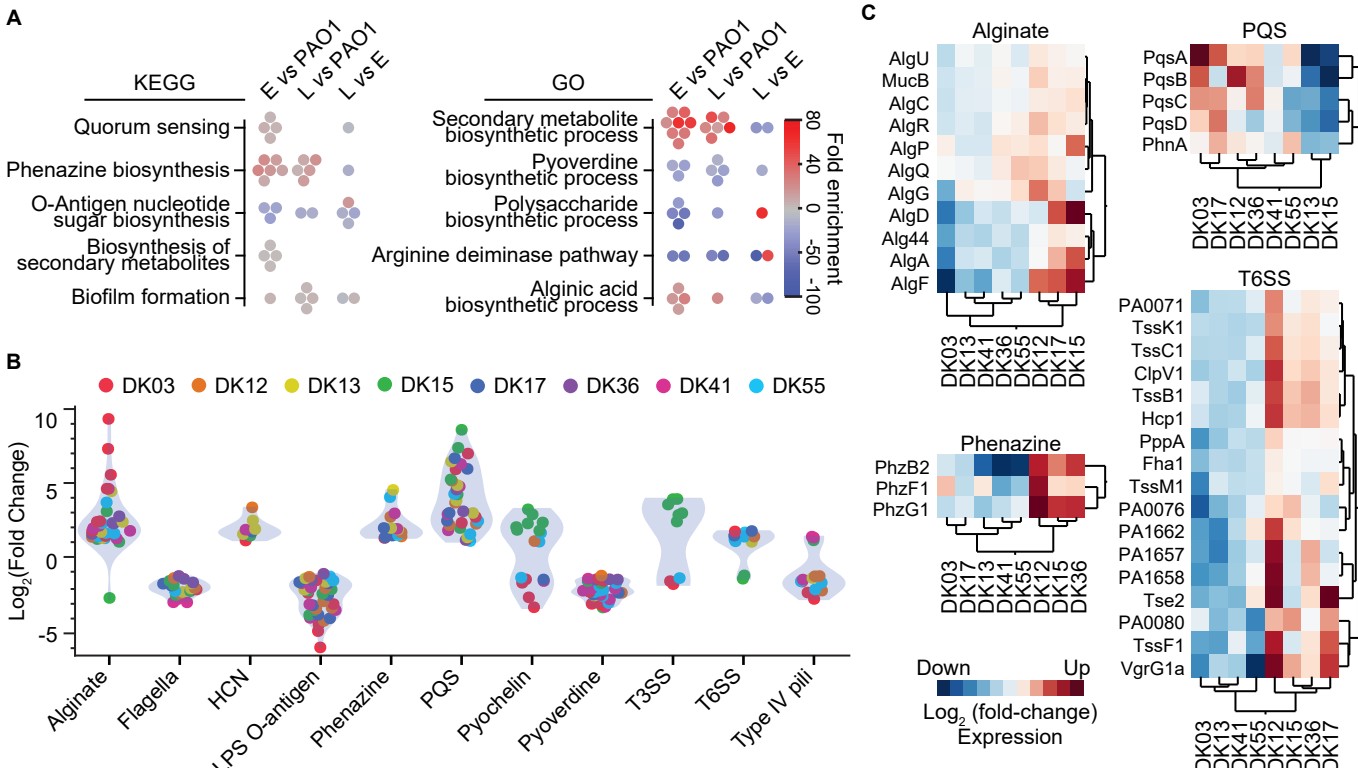

**Fig 4. Expression of virulence factors differs in clinical strains vs. PAO1.** (A) Enrichment analysis for KEGG and GO terms by the specific comparisons (Early vs. PAO1; Late vs. PAO1; Late vs. Early). Icons represent individual lineages and are colored by fold-enrichment. (B) Differential expression of virulence factors in early clinical strains vs. PAO1. Only differentially expressed proteins are represented and are colored by clone type. (C) HCA of differentially expressed proteins of virulence factors in late vs. early clinical strains. The data underlying this figure can be found in S2 Data. HCA, hierarchical cluster analysis.

metabolic specialization of late isolates. In particular, we focused on those genes encoding the phospho*enol*pyruvate-pyruvate-oxaloacetate node (S2 Fig), since such mutations might be expected to lead to elevated pyruvate secretion. This led to the identification of mutations in genes encoding the PDHc, *aceE* and *aceF*, in late strains of DK12 and DK36, respectively. The PDHc catalyzes the conversion of pyruvate to acetyl-CoA, connecting sugar metabolism with the TCA cycle (S2 Fig). Interestingly, the *aceE* and *aceF* genes are candidate pathoadaptive genes, suggesting positive selection for such mutations in pwCF [39]. In our strain collection of 474 isolates from 34 pwCF [39], we identified the presence of 18 different and independent *aceE*/*aceF* mutations (4 indels and 14 SNPs, of which 3 were synonymous) in 18 separate lineages. These lineages were present in more than half of the patients, supporting the hypothesis that modulation of the PDHc might be selected for during airway infections (S5 Fig). Moreover, in situ expression of the *aceE* and *aceF* genes is reduced in sputum samples collected from chronically infected pwCF [47]. This suggests an undescribed role of the PDHc activity which connects central carbon metabolism and pathogenicity during infection.

The DK12_L and DK36_L strains contained a +TCCC duplication at position 813 in *aceF* and a T→C transition at position 551 in *aceE*, respectively (Fig 5A). To evaluate the contribution of these mutations to the bacterial phenotype, independent of the underlying historical contingency of the clinical strains, we generated recombinant PAO1 derivative strains containing the same mutations. The *aceF* mutation leads to a frameshift starting from Lys 273, whereas the *aceE* mutation leads to a Phe→Ser amino acid change at position 184. The recombinant *aceE* and *aceF* mutant strains show reduced growth rate (Fig 5B) and increased

secretion of pyruvate (maximal pyruvate accumulation by the *aceE* mutant was 0.5 mM, whereas for *aceF*, the value was 13 mM) (Fig 5C). By comparison, maximal pyruvate accumulation by the DK12_L and DK36_L late strains was 8.5 mM and 4.6 mM, respectively (Fig 2D). The PAO1-derived *aceE* mutant strain clearly has a milder phenotype (a smaller reduction in growth rate and lower pyruvate secretion) compared with the *aceF* mutant strain, suggesting partial functionality of the PDHc. As previously reported, in *P. aeruginosa*, acetate is catabolized into acetyl-CoA, and can, therefore, metabolically complement the growth defects of PDH mutants [31,48,49]. Indeed, we were able to fully restore the growth phenotype of the *aceE* strain by supplementing the bacterial culture with acetate to replenish the pool of acetyl-CoA, whereas this only partially restored growth in the *aceF* mutant strain (Fig 5B). A similar result was obtained by expressing a wild-type copy of the *aceE* gene under its native promoter which complements the growth defect of the strain (S6A Fig). These data confirm that the frameshift in *aceF* has a much greater impact on PDHc activity than the SNP in *aceE*. Surprisingly, in laboratory conditions, phenotypes such as biofilm formation, motility, redox susceptibility, pyoverdine production, and antibiotic susceptibility showed no statistical difference between the PAO1 wild-type strain and the *aceE* and *aceF* mutant strains, except for a slight decrease in twitching motility and mildly increased tobramycin susceptibility (S6 Fig).

To characterize the effect of the *aceE* and *aceF* mutations more broadly, we performed whole cell proteomics on the *aceE* and *aceF* recombinant strains (and on the PAO1 progenitor) in presence and absence of acetate. In total, we quantified 3,246 proteins and identified 449 as differentially expressed in at least 1 comparison of either the mutant versus wild-type or of the mutant in presence versus absence of acetate (S3 Data). In the absence of acetate, the proteome of the *aceE* and *aceF* mutants was clearly different compared with that of the wild-type progenitor, PAO1 (Fig 5D). PC1, which encompasses >50% of the variance in the data set, separated the *aceF* mutant proteome from that of the wild type, whereas the proteome of the *aceE* mutant was more similar to that of the wild type, separating along PC2 (accounting for just 20% of the total variance in the data set, Fig 5D). As previously noted, supplementation of the growth medium with acetate altered both the *aceE* and *aceF* mutant proteomes, with both moving closer to that of the wild type (Fig 5D). This effect was greater in the *aceF* mutant, where growth in acetate decreased the number of up-regulated proteins (*cf*. PAO1) by more than half (from 301 to 126) (Fig 5E). Unsurprisingly, the expression of several enzymes involved in pyruvate and acetyl-CoA metabolism (e.g., lactate dehydrogenase (41), acetyl-coenzyme A synthetase (6), citrate synthase (28)) return to wild-type levels following the addition of acetate (Figs 5F and S7A). Similarly, for the clinical isolates DK12_L and DK36_L, the categories of proteins involved in amino acid/lipid metabolism and energy conversion presented the largest number of differentially expressed proteins (S7B Fig). Moreover, proteins involved in terpene, propionate, isoprenoid and branched-chain amino acid metabolism (all of which are directly connected to pyruvate and acetyl-CoA metabolism) were statistically enriched in the PAO1-derived *aceF* mutant, indicating a reorganization of both central and peripherical pathways to cope with reduced synthesis of acetyl-CoA (S7C Fig). The PAO1-derived *aceE* mutant also shows an extreme down-regulation of the oxygen-sensing transcriptional regulator Dnr, which is known to be required for denitrification but also regulates acetate metabolism and the T6SS [50–52]. This mirrors the down-regulation seen specifically for cluster B of the clinical isolates (Fig 3C), which may suggest a role for Dnr down-regulation in metabolic specialization (S7D Fig).

Importantly, the *aceE* and *aceF* mutants showed increased expression of proteins involved in alginate production and T6SS, as well as decreased expression of proteins involved in the T3SS, including the secreted factor ExoT (in strains *aceF* and DK12_L) (Fig 5G and S3 Data). This indicates that the *aceE* and *aceF* mutations contribute to the expression profile of

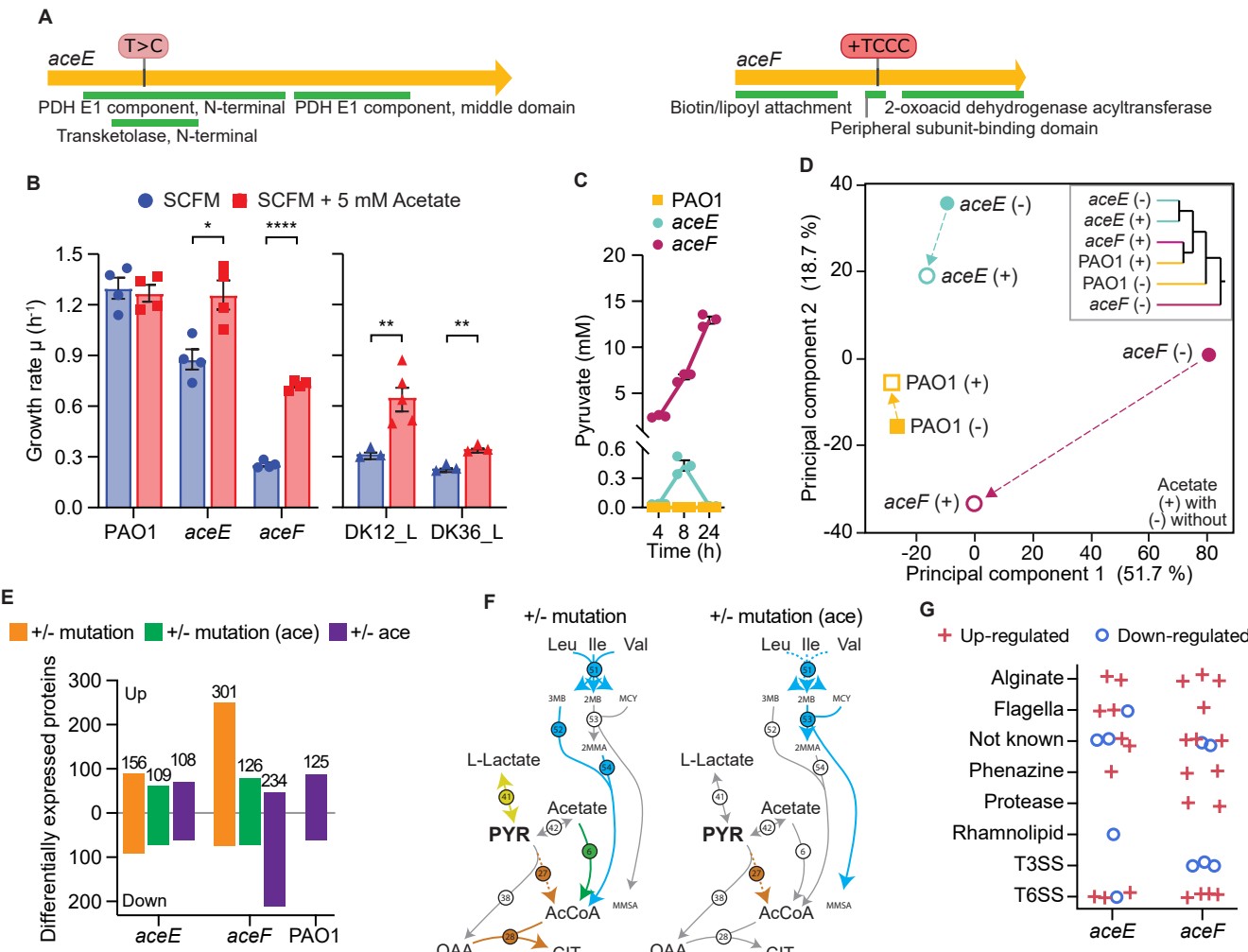

**Fig 5. *aceE* and *aceF* mutations cause metabolic specialization.** (A) Schematic of the *aceE* and *aceF* genes including location and type of mutation found in the DK36 (*aceE*) and DK12 (*aceF*) late strains. (B) Growth rate (hour$^{-1}$) in SCFM2 (blue) and SCFM2 supplemented with 5 mM acetate (red) for PAO1, *aceE* and *aceF* mutant strains, and DK12 and DK36 late clinical isolates. Bars indicate mean ± SEM, with icons representing biological replicates. Statistical significance is assessed by two-tailed unpaired parametric Welch *t* test and indicated as * ($p < 0.05$), ** ($p < 0.01$), or **** ($p < 0.0001$). (C) Pyruvate secretion (mM) for PAO1 wt (yellow), *aceE* (cyan), and *aceF* (magenta) mutant strains over 24 h. Icons represent biological replicates. (D) PCA and HCA of whole-cell proteomics for PAO1 wt and *aceE* and *aceF* mutant strains. Filled icons indicate without (−) and unfilled icons indicate with (+) 5 mM acetate. E Number of differentially expressed proteins for mutant strain vs. PAO1 wt comparisons: in orange in absence of acetate (+/− mutation) and in green in presence of acetate (+/− mutation (ace)). In purple, the number of differentially expressed proteins for either the PAO1 wt or the *aceE* or *aceF* mutant strain in presence vs. absence of acetate (+/− acetate). (F) Metabolic map of enzymes related to pyruvate and acetyl-CoA metabolism for mutant strains vs. PAO1 in absence (left) or presence (right) of acetate. Reactions are colored by their pathway if the underlining enzyme is differentially expressed (dashed if down-regulated or continuous if up-regulated). Enzymes responsible for each reaction are indicated by numbered circles. For details on individual enzymes, see S7 Fig. (G) Differentially expressed proteins involved in virulence in *aceE* and *aceF* mutant strains vs. PAO1 in absence of acetate. Jittered icons indicate specific proteins that are up-regulated (red plus) or down-regulated (blue circle). The data underlying this figure can be found in S3 and S4 Data. HCA, hierarchical cluster analysis; PCA, principal component analysis; SCFM2, synthetic cystic fibrosis medium 2.

virulence determinants shown by the late clinical isolates of DK12 and DK36 (Figs 4 and 5G and S3 Data).

## Pyruvate dehydrogenase mutations modulate pathogenicity in ALI culture infections

To test whether the secretion of pyruvate and the reduced expression of virulence determinants in the PAO1-derived *aceF* and *aceE* mutants lead to reduced infectivity, we performed

host-bacteria infections using an ALI infection model system. This model system is composed of mucociliated differentiated airway epithelial cells which represent the airways and provides insights into the host response including epithelial damage and recruitment of the immune system. Overall, the *aceF* mutant displayed a broad suppression of virulence, including reduced epithelium damage and innate immune recognition during the infection (Fig 6). The transepithelial electrical resistance (TEER) which quantifies the integrity and permeability of the epithelial layer, LDH release which quantifies the epithelium cellular damage, and the bacterial count which quantifies the growth and penetration of the bacteria through the epithelium to the basolateral side of the ALI transwells, all showed reduced values after 14 h of infection by the *aceF* mutant, compared with the PAO1 progenitor (Fig 6A). This is in line with the behavior of a Δ*pscC* mutant defective in T3SS, which also shows severely reduced virulence (Fig 6A). By contrast, the *aceE* mutant elicited epithelial damage similar to PAO1, indicating that the mutation does not influence bacterial penetration (Fig 6A). This is consistent

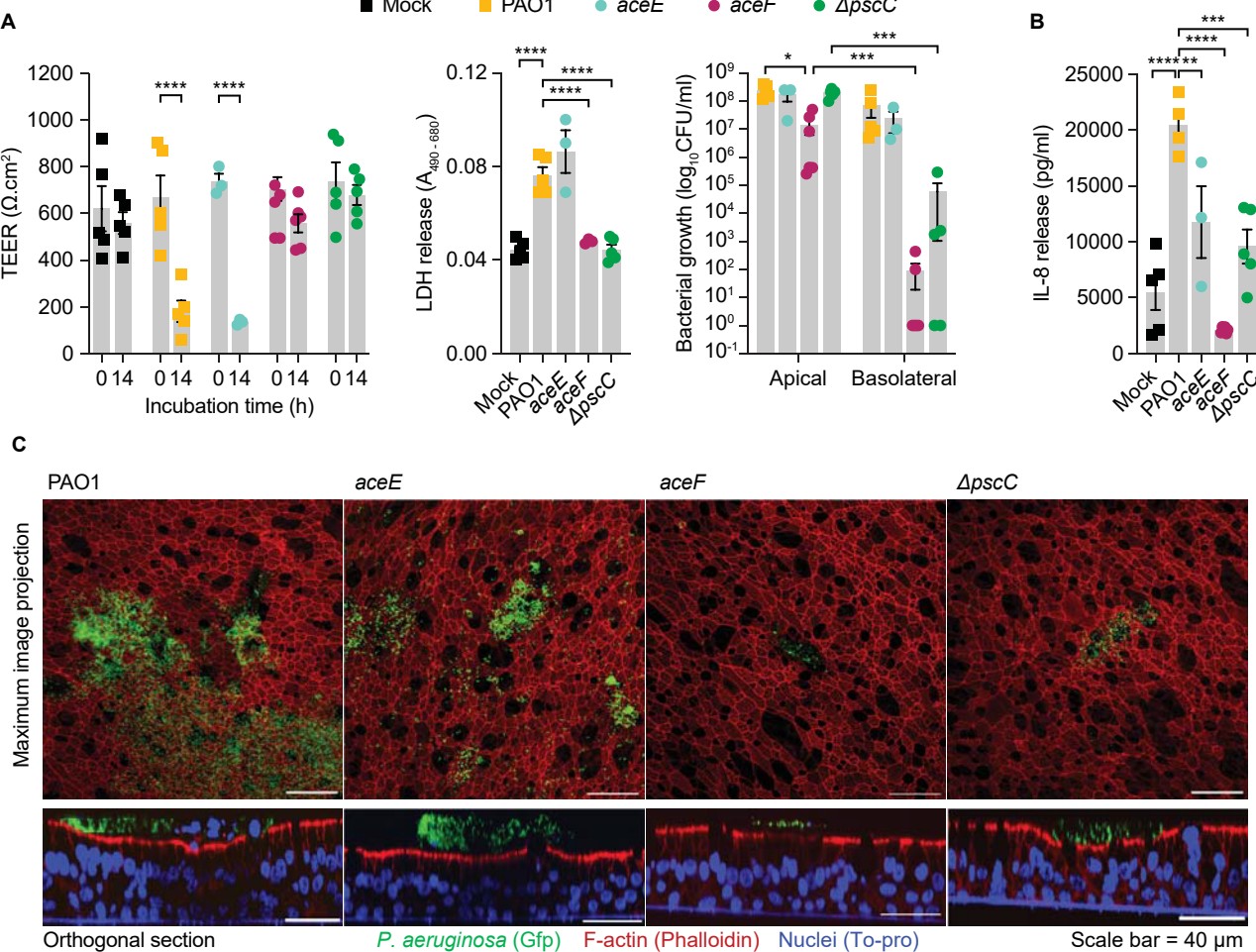

**Fig 6. Aggressiveness of *aceE* and *aceF* mutants in ALI airway infections.** (A) Mean ± SEM of TEER (Ω·cm²), LDH release, and CFUs in the apical and basolateral ALI compartments after 14 h of infection in fully differentiated BCi-NS1.1 cells. (B) IL-8 release into the basolateral media indicating inflammation caused by the invading bacteria. Icons represent each biological replicates. Mock represents un-infected control cells. Statistical significance was determined by two-way ANOVA for TEER and CFU measurements and one-way ANOVA for LDH and IL-8 measurements and indicated as * ($p < 0.05$), ** ($p < 0.01$), *** ($p < 0.001$), and **** ($p < 0.0001$). (C) Confocal images of ALI transwells following infection with *P. aeruginosa* in green (GFP), epithelium in red (Phalloidin), and nuclei in blue (To-pro). Scale bar = 40 μm. The data underlying this figure can be found in S4 Data. ALI, air–liquid interface; CFU, colony-forming unit; TEER, transepithelial electrical resistance.

with the proteomic analysis showing no significant differential expression in this strain of proteins involved in the T3SS, which is the leading cause of epithelial damage (Fig 5G). However, both the *aceE* and *aceF* mutants, together with the Δ*pscC* mutant, elicit lower interleukin 8 (IL-8) release, a cytokine that is secreted by epithelial cells and is necessary for the recruitment of the immune system at the site of infection (Fig 6B). These results are corroborated by confocal microscopy data, which show a similar colonization profile on the airway epithelium of the wild-type and *aceE* mutant, and of the *aceF* and Δ*pscC* mutant, respectively. It is worth noting that nuclei shedding, seen as nuclei (in blue) being pushed to the apical side of the epithelium (in red), was observed only for PAO1 and not for the *aceE* mutant (Fig 6C), which may suggest a slight reduction in the aggressiveness of the infection. This mechanism of reduced cellular damage and reduced inflammation seems not to depend on the growth defect associated with the *aceF* mutant, since the Δ*pscC* mutant shows comparable aggressiveness to the *aceF* mutant but a wild-type–like growth rate.

Importantly, in our longitudinal collection of clinical *P. aeruginosa* strains, 18 different lineages (specific clone types obtained from different pwCF; 89 isolates) harbored mutations in the *aceE* and *aceF* genes and/or carried additional mutations in T3SS genes [39]. In 9 lineages, PDHc mutations are present alone or anticipate additional T3SS mutations. In 7 lineages, T3SS mutations precede PDHc mutation or both systems are mutated at the same time (4 lineages) already in the first isolate of the lineage (S8 Fig).

Altogether, these results suggest a role for the *aceE* and *aceF* mutations in persistent infections and a link between pyruvate metabolism and virulence.

## Discussion

Metabolism constitutes a central process which is specifically regulated and optimized according to the nutritional resources available and the phenotypic requirements of the cell. Several studies have shown that metabolic specialization occurs in clinical isolates of *P. aeruginosa* [5,10,24,25,53]. However, little is known about whether this metabolic specialization is simply an accommodation for other significant fitness gains, i.e., metabolic compensation for phenotypes with high fitness cost, or whether it is specifically selected to enable increased persistence in the host.

By analyzing a range of longitudinal clinical isolates of *P. aeruginosa*, we found that all infection lineages underwent a substantial metabolic rewiring. This includes CF-specific patterns of carbon source (amino acids, sugars, and organic acids) assimilation and fermentation (acetate secretion). These metabolic changes, in turn, are associated with changes in proteome allocation potentially driven by mutations in metabolic genes. Adapted strains harboring mutations in genes (*aceE* and *aceF*) encoding the PDHc are characterized by a chronic-like phenotype (T3SS down-regulation, up-regulation of the T6SS, phenazines, the alginate biosynthetic pathway, and slow growth) suggesting that these mutations confer a selective advantage in the host [4]. Moreover, adapted strains secrete large amounts of pyruvate, which is an important mediator of inflammation in the host [37]. Reconstitution of clinical *aceE* and *aceF* mutations in a defined laboratory strain (PAO1) revealed that these mutations elicit reduced stimulation of the immune system and reduced epithelial layer penetration when compared with the isogenic wild-type progenitor strain (PAO1), which exhibits a highly virulent phenotype similar to that associated with acute infection. Importantly, this phenotype is only evident in the ALI infection model system and not in standard laboratory conditions since virulence and pathogenicity are only elicited in the presence of the host. These findings are consistent with the notion that strains with reduced virulence are selected during within-patient evolution in pwCF to reduce host tissue damage and limit the activity of the immune system [54].

Such reduced aggressiveness in the *aceF* mutant appears to be mediated by lower expression of the T3SS effector, ExoT, which induces apoptosis in host cells [16,17]. This is consistent with previous studies showing that mutations in the PDHc-encoding genes lead to repression of T3S in *P. aeruginosa* [31,55]. Mutations in the enzyme *iso*citrate lyase, involved in the glyoxylate shunt of the TCA cycle, have also been associated with reduced T3SS activity, suggestive of a direct correlation of metabolism to virulence [29]. This is the case for the DK15 late isolate in our collection, which shows reduced expression of *iso*citrate lyase together with suppression of the ExoT exotoxin. Given the tendency in our collection for *aceE* and *aceF* mutations to precede mutations in the T3SS-encoding genes, it is interesting to consider if mutations in the PDHc and/or modified PDHc activity could play a broader role in promoting a persistence phenotype during evolution in the host by reducing T3SS-dependent virulence. However, further investigation would be needed to establish such a mechanism. Nevertheless, the diversity of *aceE* and *aceF* mutations indicate that the activity of the PDHc can be fine-tuned accordingly to the specific host environment and consequently lead to different infection outcomes. Indeed, while the *aceE* mutant (carrying just an SNP) shows comparable aggressiveness to PAO1 but reduced recruitment of the immune system (IL-8 secretion), the *aceF* mutant (carrying a frameshifting indel) combines the effect of the reduced aggressiveness and that of the reduced immune recruitment.

It is worth noting that pyruvate, in addition to its crucial role as intermediate of central carbon metabolism, is a potent scavenger of ROS which use has been suggested as an anti-inflammatory treatment in several diseases [36,56]. Due to the sustained activation of neutrophils and other defects in the homeostatic processes of pulmonary epithelia in CF airways, an abnormal flux of ROS is present which can be alleviated by the bacterial pyruvate secretion [57]. During exacerbation events, the concentration of pyruvate increases (0.3 mM) which might counterbalance the increased inflammation [58]. Indeed, it has been shown that treatment of human lung epithelial cells with pyruvate reduces the inflammatory response by reducing the production of IL-8 [59,60]. It induces a potent shutdown of the response to LPS in dendritic cells [61] and reduces alveolar tissue destruction in a COPD model system [62]. Similarly, inhibition of the PDHc and redirection of the pyruvate flux has been suggested both as a potential anti-inflammation therapy for chronic metabolic diseases [63] and as a cancer treatment [64]. Moreover, fine-tuning of the flux through the mitochondrial PDHc in macrophages and dendritic cells is key for regulating their polarization and thus the balancing between pro- and anti-inflammatory responses [38,65]. LPS-induced polarization of macrophages is prevented by pharmacological inhibition of pyruvate import into the mitochondria [66]. Moreover, flux through the PDHc, possibly altered by the pyruvate secreted by the *aceE* and *aceF* mutants, controls the production of the antibacterial immunometabolite, itaconate, by macrophages [67–70]. Therefore, the reduced recognition from the epithelial layer of the *aceE* and *aceF* mutants in this study may depend on the secretion of bacterial pyruvate, which is known to mediate an important inflammatory response in both the epithelium and the immune cells [37,38,56].

The metabolic specialization and specifically the PDHc dysregulation seen in the clinical isolates may, therefore, serve as a mechanism for ensuring active pyruvate secretion and suppression of the immune system by a mechanism of cross-feeding. The evolutionary benefit of such a mechanism of host–pathogen crosstalk during persistent infections may explain why, in many cases, ROS produced in the CF airways appear to select for metabolic specialization in *P. aeruginosa* [71]. Moreover, the consequent phenotypic change of metabolically adapted strains limits epithelial damage to avoid any further inflammation. Indeed, changes in the T3SS, the T6SS, phenazines, alginate biosynthesis, and growth physiology are all hallmarks of a transition from an acute (high virulence) to a chronic (low virulence) phenotype which are specifically

selected for during colonization of pwCF [4,54]. These combined mechanisms of cross-feeding and reduced virulence in turn promote persistence in the host by a change in the infection microenvironment which can benefit the entire population of infecting bacteria. Moreover, they can provide a rationale for the broad accumulation and maintenance of PDHc mutations in *P. aeruginosa* clinical CF isolates and the reduced expression of the PDHc in sputum samples [39,47]. This suggests a highly complex and near universal role of the pyruvate node in regulation of both host and pathogen (and their respective functions in vivo) through a conserved central metabolite. Such a mechanism might be engaged years before the development of antibiotic resistance, because of its high relevance for the establishment of a chronic phenotype.

The mechanism of metabolic specialization presented here is only one of many different mechanisms that *P. aeruginosa* might use for long-term survival in the host. However, the connection between metabolic behaviors and virulence phenotype (acute or chronic) are still poorly characterized, limiting our capacity to design treatments that counteract such processes. While our work focuses on a limited collection of clinical strains, a systematic analysis of the heterogeneous populations infecting pwCF is necessary to fully evaluate the role of metabolic specialization in persistence. Designing new methods, e.g., for high-throughput metabolomics analyses under in vivo-like conditions, such as those recreated by ALI or organoid cultures [71] which go beyond the limitation of laboratory conditions, will allow the characterization of new persistence mechanisms by clarifying the relationship between the host, the pathogen, and other microbial species of the host microbiota. Indeed, laboratory conditions and classical phenotype screening approaches are inadequate to unravel complex host–pathogen interactions. Still, it is important to note that the pyruvate secretion and the associated phenotypic changes described here might be beneficial not only for the PDHc mutated population but also for other nearby "wild-type" cells, since they alleviate the airway inflammation and limit the recruitment of the immune system. Importantly, metabolic specialization might be equally relevant in other persistent infections such as those caused by *Escherichia coli*, *Staphylococcus aureus*, and *Mycobacterium tuberculosis* which are similarly threatening and difficult to tackle [2]. Finally, understanding specific mechanisms linking metabolism, energy balance and virulence, and most importantly how the relationship between the host and the pathogen changes during an infection, could provide new opportunities for more efficient and/or complimentary treatments, beyond the classical antibiotic treatment, which are greatly needed.

## Methods

### Strains and media

The collection of clinical isolates used in this study is a subset of the collection published in Marvig and colleagues and Bartell and colleagues [9,39]. The local ethics committee at the Capital Region of Denmark (Region Hovedstaden) approved the use of the stored *P. aeruginosa* isolates (registration number H-21078844). For information about specific clinical isolates, see S1 Table.

For construction of *aceE*, *aceF*, and Δ*pscC* mutant strains, derivatives of pACRISPR were constructed with the Uracil Specific Excision Reagent (USER) cloning [72]. The plasmids and primers used are listed in S1 and S2 Tables, respectively. Target DNA fragments were amplified from gDNA of clinical strains DK12 Late and DK36 Late using the Phusion U polymerase kit (Thermo Fisher Scientific, United States of America). For the Δ*pscC* mutant strain primers were designed to create matching short overlaps between the DNA fragments that should be stitched together in the final plasmid. The resulting products were treated with FastDigest

DpnI enzyme (Thermo Fisher Scientific, USA) and ligated with the USER Enzyme (New England Biolabs, USA). PCR was performed on random *E. coli* DH5alpha colonies using One-Taq 2X Master Mix (New England Biolabs, USA) to confirm the correct insertions and further sequenced with Sanger method (Eurofins Scientific, Luxembourg). *aceE*, *aceF*, and Δ*pscC* mutations in the genome of *P. aeruginosa* PAO1 were introduced using the previously developed CRISPR-Cas9 system and following the indicated protocol [73] with the addition of 0.2% arabinose in the growth medium during transformation. To confirm the presence of desired mutations, PCR fragment of the genome around the mutations was amplified and sequenced with Sanger sequencing (Eurofins Scientific, Luxembourg). To perform confocal microscopy analyses, strains were tagged with GFP using 4 parental mating using a mini-Tn7 delivery method [74] but for strains *aceE* and *aceF* with the additional supplementation of 5 mM acetate to ensure growth of PDHc dysregulated target strains. Transformants were identified by green fluorescence and validated by comparing growth rate to the untagged target strain.

To complement the mutation in the PDH (*aceE* gene), we constructed a complementation strain *aceE(rev)* containing a wild-type copy of the *aceE* gene under the control of its native promoter and integrated it into the genome of the *aceE* mutant using the mini-Tn7 delivery method [74] as above. The plasmid pIP281 carrying the *aceE* gene and the Tn7 system was assembled with USER cloning [72]. The plasmids and primers used are listed in S1 and S2 Tables, respectively.

Bacteria were grown in Synthetic Cystic Fibrosis Media 2 (SCFM2) [75]. To reduce viscosity and allow for HPLC analysis, DNA and mucins were excluded [76]. Cultures were grown at 37°C and 250 rpm.

## Dynamic exo-metabolomics analyses

Sampling of supernatants was performed in 96-well deep well plates (Cat. No. 0030502302; Eppendorf, Hamburg, Germany) with an air:liquid ratio of 1:1 using a high-throughput dilution-based growth method as previously described [77]. Immediately before sampling, $OD_{600}$ of the cultures were measured in a Synergy MX microtiter plate reader (BioTek Instruments, Winooski, Vermont, USA). Supernatants were stored at −80°C until HPLC-analysis. For organic acids and sugars (glucose, lactate, formate, acetate, and pyruvate), a Dionex Ultimate 3000 system (Thermo Scientific, Waltham, USA) with a HPx87H ion exclusion column (125–0140, Aminex, Dublin, Ireland), equipped with a guard column (125–0129, BioRad, Hercules, California, USA) and guard column holder (125–0131, BioRad, Hercules, California, USA) was used. Samples were injected with an injection volume of 20 µl and eluted, using a 5 mM $H_2SO_4$ mobile phase, at an isocratic flow of 0.6 ml min$^{-1}$ at 45°C for 30 min. Pyruvate was analyzed by UV detection at a wavelength of 210 nm, using a System Gold 166 UV-detector (Beckman Coulter, Brea, USA), while the rest of the metabolites were analyzed by RI detection, using a Smartline RI detector 2300 (KNAUER Wissenschaftliche Geräte, Berlin, Germany). For amino acids (aspartic acid, glutamic acid, serine, histidine, glycine, threonine, arginine, alanine, tyrosine, valine, phenylalanine, isoleucine, leucine, lysine, and proline), 20 µl of the thawed sample was diluted 1:10 by mixing with 80 µl Ultrapure MilliQ water and 100 µl of internal standard (20 µg/ml 2-aminobutyric acid and sarcosine) before injection of 56.5 µl into the instrument. Prior to injection into the column, derivatization of amino acids was performed in the HPLC-instrument by automatic mixing with the following eluents: (i) 0.5% (v/v) 3-mercaptopropionic acid in borate buffer 0.4 M at pH 10.2; (ii) 120 mM iodoacetic acid in 140 mM NaOH; (iii) OPA reagent (10 mg/ml *o*-phtalaldehyde and 3-mercaptopropionic acid in 0.4 M borate buffer); (iv) FMOC reagent (2.5 mg/ml 9-fluorenylmethyl chloroformate in acetonitrile); and (v) buffer A (40 mM $Na_2HPO_4$, 0.02% (w/v) $NaN_3$ at pH = 7). Following

derivatization, the samples were separated isocratically on a Dionex Ultimater 3000 HPLC with fluorescence detector (Thermo Scientific, Waltham, USA) through a Gemini C18 column (00F-4439-E0, Phenomenex, Værløse, Denmark) equipped with a SecurityGuard Gemini C18 guard column (AJ0-7597, Phenomenex, Værløse, Denmark) with 5 mM $H_2SO_4$ at a flowrate of 1 ml min$^{-1}$ at 37˚C for 31 min. Amino acids were detected using an UltiMate 3000 Fluorescence variable wavelength UV detector (FLD-3400RS, Waltham, Massachusetts, USA).

## Proteomic analyses

To maintain a 1:1 air:liquid ratio, as in the metabolomic analysis, 25 ml SCFM2 cultures were inoculated from ON cultures in 50 ml Falcon tubes at $OD_{600}$ of 0.05 and then incubated in an orbital shaker at 37˚C and 250 rpm until sampling by centrifugation at mid-exponential phase. Five biological replicates were analyzed for PAO1 and the clinical isolates while 3 biological replicates for the PAO1 derivative strains *aceE* and *aceF* both in presence and absence of 5 mM acetate. Pellets were washed twice with PBS and stored at −80˚C until protein extraction. No specific preparation or enrichments were performed to identify secreted proteins. For experiment with clinical isolates, protein extraction was done in Lysis buffer A (100 mM Tris, 50 mM NaCl, 1 mM tris(2-carboxyethyl)phosphine (TCEP), 10% glycerol, pH = 7.5) with *cOmplete Mini protease inhibitor* (Roche) by sonication at amplitude 10 for $3 \times 10$ s cycles with 20 s cooling between, followed by acetone precipitation and resuspension in Lysis buffer B (6 M Guanidinium hydrochloride, 5 mM TCEP, 10 mM chloroacetamide, 100 mM Tris-HCl, pH = 8.5). For experiment with PAO1 derivative mutant strains, protein extraction was done in Lysis buffer B by bead beating with 3-mm zirconium oxide beads at 99˚C for 5 min in a Tissuelyzer (Retsch, MM 400) at 25 Hz, then boiled, still at 99˚C, in heat block (Eppendorf Thermomixer C) for 10 min while shaking/mixing at 2,000 rpm. In both cases, protein concentrations were determined by micro BCA Protein Assay Kit (Thermo Scientific, prod #23235) and 100 μg protein was used for trypsin digest in Trypsin/Lys-C Buffer (0.1 μg/μl trypsin, 50 mM Ammonium Bicarbonate). The reaction was stopped by addition of 10 μl 10% TFA and samples were desalted by stagetipping with C18 filter plugs (Solaμ HRP 2 mg/1 ml 96-well plate, Thermo Scientific). Peptide samples were stored in 40 μl of 0.1% formic acid at 4˚C until LC-MS analysis.

For the experiment with clinical isolates, LC-MS/MS was carried out using a CapLC system (Thermo Fisher Scientific, Waltham, USA) coupled to an Orbitrap Q-exactive HF-X mass spectrometer (Thermo Fisher Scientific, Waltham, USA). The first samples were captured at a flow of 10 μl/min on a precolumn (μ-pre-column C18 PepMap 100, 5 μm, 100 Å) and then at a flow of 1.2 μl/min. Peptides were separated on a 15 cm C18 easy spray column (PepMap RSLC C18 2 μm, 100Å, 150 μm × 15 cm). The applied gradient went from 4% acetonitrile in water to 76% over a total of 60 min. MS-level scans were performed with Orbitrap resolution set to 60,000; AGC Target 3.0e6; maximum injection time 50 ms; intensity threshold 5.0e3; dynamic exclusion 25 s. Data-dependent MS2 selection was performed in Top 20 Speed mode with HCD collision energy set to 28% (AGC target 1.0e4, maximum injection time 22 ms, Isolation window 1.2 m/z). For the experiment with the *aceE* and *aceF* mutants, peptides were loaded onto a 2 cm C18 trap column (Thermo Fisher 164946), connected in-line to a 15 cm C18 reverse-phase analytical column (Thermo EasySpray ES904) using 100% Buffer A (0.1% formic acid in water) at 750 bar, using the Thermo EasyLC 1200 HPLC system, and the column oven operating at 30˚C. Peptides were eluted over a 70 min gradient ranging from 10% to 60% of 80% acetonitrile, 0.1% formic acid at 250 nL/min, and the Orbitrap Exploris instrument (Thermo Fisher Scientific) was run in DIA mode with FAIMS Pro Interface (Thermo Fisher Scientific) with CV of −45 V. Full MS spectra were collected at a resolution of 120,000, with an

AGC target of 300% or maximum injection time set to "auto" and a scan range of 400 to 1,000 m/z. The MS2 spectra were obtained in DIA mode in the orbitrap operating at a resolution of 60.000, with an AGC target 1,000% or maximum injection time set to "auto," and a normalized HCD collision energy of 32. The isolation window was set to 6 m/z with a 1 m/z overlap and window placement on. Each DIA experiment covered a range of 200 m/z resulting in 3 DIA experiments (400 to 600 m/z, 600 to 800 m/z, and 800 to 1,000 m/z). Between the DIA experiments a full MS scan is performed. MS performance was verified for consistency by running complex cell lysate quality control standards, and chromatography was monitored to check for reproducibility.

## Phenotypic characterizations

Growth curves were performed by inoculating 1 μl of overnight culture in 149 μl of media, using 96-well microtiter plates (Cat. No. 650001; Greiner Bio-One, Kremsmünster, Austria), covered with plate seals (Ref. 4306311, Thermo Fisher Scientific, United Kingdom) and incubated at 37˚C and 250 rpm in a BioTek ELx808 Absorbance Microtiter Reader (BioTek Instruments, Winooski, Vermont, USA) for 24 to 48 h. Antibiotic MICs were determined by microdilution. ON cultures were standardized to $OD_{600} = 0.5$ and diluted 1:2,500 to reach $5 \times 10^5$ CFU/ml in fresh SCFM2 media (LB for Azithromycin). Growth assays were performed at increasing antibiotic concentrations and MIC determined based on final OD. Resistance to oxidative stress was measured as the diameter of clearance zones around diffusion disks saturated with 5 μl of fresh 30% $H_2O_2$ after 24 h incubation at 37˚C on LB agar plates cast with 3 ml overlay agar containing 100 μl of LB ON culture standardized to $OD_{600} = 1$. Pyoverdine production was measured as the fluorescence at 400/460 nm excitation/emission on a Synergy H1 Hybrid Multi-Mode Reader (BioTek Instruments, Winooski, Vermont, USA) of supernatants normalized against $OD_{600}$ of ON cultures in King's B medium. Biofilm formation assay was done as previously described for NUNC peg lids [9] (NUNC cat no. 445497). Motility was measured as the diameter of the motility zone around single colonies deposited in middle layer of 0.3% (swimming), surface layer of 0.6% (swarming), or bottom layer of 1.5% (twitching) LB agar motility plates, after incubation at 37˚C for 24 to 48 h. For PAO1 wild-type and *aceEF* mutant strains, pyruvate secretion was determined from supernatants of samples taken after 0, 4, 8, and 24 h of growth in SCFM2. Supernatants were stored at −80˚C and analyzed by the HPLC method also used for exo-metabolomics.

## Infection of ALI cultures

For the ALI infections, the BCi-NS1.1 cells were used [78]. Cells were cultured in Pneumacult-Ex Plus medium (STEMCELL Technologies, 05040) supplemented with Pneumacult-Ex 50x supplement (STEMCELL Technologies, 05008), 96 ng/ml hydrocortisone (STEMCELL Technologies, 07925), and 10 μm Y-27632 ROCK inhibitor (Bio-Techne #1254/10) in a 37˚C, 5% $CO_2$ humidified incubator. Following expansion, $1.5 \times 10^5$ cells were seeded onto 6.5-diameter-size transwells with 0.4 μm pore polyester membrane inserts (Corning Incorporated, 3470) previously coated with human type I collagen (Gibco, A1048301). ALI was established once cells reached full confluency by removing media from the apical chamber and replacing media in the basolateral chamber with Pneumacult-ALI maintenance medium (STEMCELL Technologies, 05001). Pneumacult-ALI maintenance medium was supplemented with Pneumacult-ALI 10× supplement (STEMCELL Technologies, 05003), Pneumacult-ALI maintenance supplement (STEMCELL Technologies, 05006), 480 ng/ml hydrocortisone, and 4 μg/ml heparin (STEMCELL Technologies, 07980). ALI cultures were grown in a 37˚C, 5% $CO_2$ humidified incubator for 30 days, with media replacement every 2 to 3 days. Epithelial polarization was

monitored by measurements of the TEER using a chopstick electrode (STX2; World Precision Instruments). Following 15 days under ALI conditions, the apical surface was washed with PBS every media change to remove accumulated mucus. Biological replicates of bacterial strains were obtained from single colonies on LB agar plates grown ON as precultures in LB and then diluted to an $OD_{600}$ of 0.05 before sampling at mid-exponential phase by centrifugation, washing with PBS and resuspending in PBS at a density of $10^5$ CFU/ml. Fully differentiated BCi-NS1.1 cells were inoculated with $10^3$ CFU from the apical side, diluted in 10 μl PBS. Control wells were incubated with bacteria-free PBS. Cells were incubated for 14 h at 37°C, followed by addition of 200 μl PBS to the apical side and measurement of the TEER. CFUs were determined by platting 10 μl of 6-fold serial dilutions on LB-agar plates in technical triplicates both for the initial inoculum, as well as for the apical and basolateral solutions following TEER measurements. The basolateral media was also used for measurements of LDH and IL-8 release in technical triplicates, using the Invitrogen CyQUANT LDH Cytotoxicity Assay Kit (Invitrogen, C20301) and Human IL-8/CXCL8 DuoSet ELISA Kit (R&D Systems, DY208) according to the manufacturer's instructions. BCi-NS1.1 cells on transwell inserts were rinsed once with PBS and fixed by adding 4% (wt/vol) paraformaldehyde (PFA) to both apical and basolateral chambers for 20 min at 4°C. After washing, cells were permeabilized and blocked for 1 h with a buffer containing 3% BSA, 1% Saponin, and 1% Triton X-100 in PBS. Cells were stained on the apical side with Phalloidin-AF488 (Invitrogen, 65-0863-14) and TO-PRO3 (Biolegend, 304008) diluted in a staining buffer (3% BSA and 1% Saponin in PBS) at a 1:500 dilution for 2 h at room temperature. Transwells were removed from their supports with a scalpel and mounted on glass slides with VECTASHIELD Antifade Mounting Medium (VWR, VECTH-1000). Images were acquired with a Carl Zeiss LSM 510 Confocal Laser Scanning Microscope (40× magnification, 1.3 oil) and analyzed using the ImageJ software.

## Data analysis

For exometabolomics, all chromatograms were analyzed and used to construct standard curves constructed for absolute quantification of concentrations for all 20 metabolites, using software Chromeleon v7.2.9. All other analysis of exo-metabolomic data was done in JMP Pro 15.0. To compare between different strains and media batch-effects, $OD_{600}$ was normalized against final $OD_{600}$ of that strain in the experiment and concentrations were normalized against specific concentrations in SCFM2 controls for each batch. Missing values were replaced with 20% of lowest value detected for any metabolite in any sample. PCA and HCA were done on $C_{metabolite}$ of all quantified metabolites for 16 samples (8 time points from biological duplicates) of each strain. PCA and iterative k-means clustering was done on covariance using JMP Pro 15.0 software. HCA was performed using the "average" clustering method and "correlation" for distance using the R package pvclust 2.2. Reliability of the clusters were analyzed by 10,000 bootstraps. Metabolite half-life ($OD_{50}$), defined as the normalized $OD_{600}$-value, where 50% of the starting concentration of a metabolite is present, was calculated from the sigmoidal mechanistic growth model (Equation: a(1 –bExp(–cx)) where a = asymptote, b = scale, and c = rate) [79]. Naïve and adapted metabotypes were designated based on PCA, iterative k-means clustering (range 3 to 10) followed by curation based on HCA. Specifically, iterative k-means clustering readily identified adapted metabotypes 2 and 3 based on 3 clusters and the highest Cubic Cluster Criterion statistical analysis. Cluster 1 is determined based on 5 k-means and agrees with the grouping generated by HCA followed by bootstrapping. Differences of metabolic preferences between metabotypes were analyzed by comparing the assimilation and secretion profiles between strains using $OD_{50}$ values. The net balance of carbon sources assimilated during the growth was calculated by summing the amount (mM concentration) of

assimilated metabolites and subtracting the secreted ones. Metabolomics data are enclosed in S1 Data.

For the experiment with clinical isolates, proteomic raw files were analyzed using Proteome discoverer 2.4 software with the following settings: Fixed modifications: Carbamidomethyl (C) and Variable modifications: oxidation of methionine residues. First search mass tolerance 20 ppm and a MS/MS tolerance of 20 ppm. Trypsin as enzyme and allowing 1 missed cleavage. FDR was set at 0.1%. The Match between runs window was set to 0.7 min. Quantification was only based on unique peptides and normalization between samples was based on total peptide amount. For the experiment with mutant strains, raw files were analyzed using Spectronaut (version 16.2). Dynamic modifications were set as Oxidation (M) and Acetyl on protein N-termini. Cysteine carbamidomethyl was set as a static modification. All results were filtered to a 1% FDR, and protein quantitation done on the MS1 level. The data was normalized by RT dependent local regression model [80] and protein groups were inferred by IDPicker. In both cases, spectra were matched against the *P. aeruginosa* PAO1 reference proteome (UniProt Proteome ID UP000002438). In both experiments, any protein that was not quantified in at least 3 of 5 or 2 of 3 biological replicates for all strains were excluded from the analysis. Using JMP Pro 15.0, abundances were normalized by $Log_2$-transformation and biological replicates used for missing value imputation. PCA Wide was done on correlations using JMP Pro 15.0 and HCA was performed on mean $Log_2$(abundance) and $Log_2$(Fold change) for the relevant comparisons. HCA was performed using the "average" clustering method and "correlation" for distance using the R package pvclust 2.2. Reliability of the clusters were analyzed by 10,000 bootstraps. Differential expression was determined by two-way ANOVA with Tukey's multiple comparisons test, using GraphPad Prism 9.3.1, and defined as those protein-comparisons where adjusted $P$ value $\leq 0.05$ and $Log_2$(Fold Change) $\geq |0.6|$. Enrichment analysis was done using the DAVID Functional Annotation Bioinformatics Microarray Analysis tool from lists of Locus Tags of proteins that were differentially expressed, separated into lists of up-regulated and down-regulated proteins, respectively, for each of the relevant strain comparisons. The reference genome was set as *Pseudomonas aeruginosa*. Proteomics data are enclosed in S2 and S3 Data.

Growth rates and maxOD were calculated in JMP Pro 15.0. Blanks were first subtracted from $OD_{600}$-measurements and values converted to $cm^{-1}$ (using Greiner dimensions for pathlength). The stationary phase was excluded, and growth rates were calculated by fitting the Exponential 3P model to the exponential phase ($r^2 > 0.99$). Mean $\pm$ SD was calculated from biological replicates. GraphPad Prism 9.3.1 was used for statistical analysis of biofilm formation, motility, redox sensitivity, pyoverdine production, and MICs (see S6 Fig for details), as well as for ALI infection experiments. ALI data are represented as mean $\pm$ SEM. Replicates represent independent experiments performed with cells from different passages. Statistical comparisons were calculated using two-way ANOVA for TEER and CFU measurements and one-way ANOVA for LDH and IL-8 measurements. Statistical significance was considered for $p$ value $< 0.05$. All figures were finalized in Adobe Illustrator Artwork 27.0.

## Supporting information

**S1 Fig.** (A) Number of differentially expressed proteins (top) and $Log_2$(Fold change) of individual differentially expressed proteins in late vs. early clinical isolates, separated by lineage and on the x-axis by COG categories. (B) Complete enrichment analysis showing fold-enrichments separated by comparison and lineage on x-axis and by KEGG and GO terms on y-axis. The data underlying this figure can be found in S2 Data.
(EPS)

**S2 Fig.** Metabolic maps showing the differences between early clinical strains vs. PAO1 (left) and late vs. early clinical strains (right) in pathways related to the catabolism of nutrients present in SCFM2 through central carbon metabolism. Arrows represent individual metabolite-conversions colored by their pathway. Reactions are colored if the underlining enzyme is differentially expressed in more than one clone type—either with dashed (down-regulated) or full (up-regulated) lines, or both. Enzymes responsible for each reaction are indicated by numbered circles. For details on individual enzymes, see S3 Fig. Transporters are shown at the bottom as arrows crossing bacterial cell membrane with their specific transported metabolites. Asp, Glu, and acetate are labeled with * to indicate that they are represented in 2 places on the map, due to their involvement in different parts of central carbon metabolism. The data underlying this figure can be found in S2 Data.
(EPS)

**S3 Fig.** (A) Metabolic map in the same style as S2 Fig for late clinical strains vs. PAO1. (B) Table of genes/Locus Tags of all metabolic enzymes included in the map. (C, D) Bar charts showing Log$_2$(Fold change) of all differentially expressed proteins highlighted on the metabolic maps. Icons colored by lineage and separated into 3 comparisons on x-axis (Early vs. PAO1; Late vs. Early; Late vs. PAO1). Panel c contains metabolic enzymes and panel d contains transporters. The data underlying this figure can be found in S2 Data.
(EPS)

**S4 Fig.** (A) Hierarchical cluster analysis in the same style as Fig 4C, showing differential expression in late vs. early clinical strains for several more virulence categories. (B) The Log$_2$(Fold change) of differentially expressed proteins in late vs. early clinical strains separated into the relevant virulence categories and further separated into the 2 main clusters from panel a. Icons are colored by lineage and lines depict the difference in mean Log$_2$(Fold change) between the 2 clusters. The data underlying this figure can be found in S2 Data.
(EPS)

**S5 Fig.** Schematic of all 18 unique mutations observed for *aceE* (top) and *aceF* (bottom) in our collection of clinical isolates of *P. aeruginosa*. Gray lines indicate mutation site and letters indicate nucleotide-sequence colored by type of mutation. The data underlying this figure can be found in Marvig and colleagues [39].
(EPS)

**S6 Fig.** (A) Complementation analysis of *aceE* mutant. Growth rate (hour$^{-1}$) in SCFM for PAO1, *aceE* mutant and complementation strain *aceE(rev)*. Bars indicate mean ± SEM, with icons representing biological replicates. Statistical significance was assessed by one-way ANOVA with Tukey's multiple comparisons test and indicated as * ($p < 0.05$). (B) Potential for biofilm formation (surface attachment) of PAO1 wt (blue), as well as *aceE* (red) and *aceF* (green) mutant strains and DK12 and DK36 late clinical strains after 24 and 48 h, respectively. For the clinical isolates, strain is indicated on the x-axes and incubation time is indicated as blue (24 h) and red (48 h). Attachment is measured as the ratio of surface-attached cells (OD$_{590}$) to total number of cells (OD$_{600}$) after incubation. For PAO1 wt and *aceE* and *aceF* mutants, all strains were compared by one-way ANOVA. For clinical isolates, each strain was compared to itself after 24 and 48 h of incubation, respectively, using two-tailed unpaired parametric Welch *t* test. Significance is indicated as "ns" ($p > 0.05$), * ($p < 0.05$), ** ($p < 0.01$), *** ($p < 0.001$). Same coloring and statistical analysis for PAO1 and mutant strains in panels B–D. (C) Motility measured as the diameter (mm) of the zone of growth in motility plates (LB agar). Swimming motility was clearly visible after 24 h of incubation. Swarming and twitching plates required 48 h of incubation. (D) Redox sensitivity measured as the diameter of the

clearance zone (mm) on bacterial lawns (LB agar) of each strain from $H_2O_2$ diffusion disks after 24 h. (E) Pyoverdine production measured as relative fluorescence ($F/OD_{600}$) of each strain after 24 h of growth in King's B medium. For panels A–D, icons indicate biological replicates and bars represent the mean ± SEM. (F) MICs of PAO1 wt and *aceE* and *aceF* mutant strains. Each cell represents the maxOD of growth curves under the given condition, following the color gradient to the right of each heatmap. The MIC is the concentration where no growth is observed (white). Each heat map shows the MIC for all 3 strains for a given antibiotic (Ceftazidime, Meropenem, Piperacillin, Azithromycin, Chloramphenicol, Ciprofloxacin, and Tobramycin). Concentrations (µg/ml) increase 2-fold downward on the vertical axis and the specific strain is given on the horizontal axis. Azithromycin MICs were determined in LB, while all other MICs were determined in SCFM2. Piperacillin was used in combination with the β-lactamase inhibitor Tazobactam. The data underlying this figure can be found in S4 Data.
(EPS)

**S7 Fig.** (A) Metabolic map in the same style as S2 Fig for *aceF* mutant strain vs. PAO1 reference in SCFM2 in absence (left) and presence (right) of acetate. (B) Parallel plot showing the number of differentially expressed proteins, separated by metabolic COG categories, in *aceE* (cyan) and *aceF* (magenta) mutant strains vs. PAO1. (C) Enrichment analysis showing fold enrichment on x-axis separated by KEGG and GO terms on y-axis for *aceE* (left) and *aceF* (right) mutant strains vs. PAO1. (D) The -Log2(Fold change) of the Dnr transcriptional regulator in late vs. early clinical isolates (same holds for late strains vs. PAO1) for lineages in Cluster A (left) and B (middle), as well as in the *aceE* mutant strain (right) when compared to PAO1 wt in SCFM2 in absence (+/− mutation) and presence (Ace) of acetate or when compared to itself in presence of acetate (+/− Acetate). The data underlying this figure can be found in S3 Data.
(EPS)

**S8 Fig. Longitudinal collection of *P. aeruginosa* clinical isolates from CF airways.** Icons indicate isolates carrying mutations in genes encoding PDHc (red squares) and/or T3SS (purple triangles) proteins, as well as isolates with no mutations in either (gray dots). Isolates are separated by Clone Type and further separated by patient-specific lineages within Clone Types (A, B, C). The x-axis shows the length of infection (years) since the first isolate of the given lineage. The data underlying this figure can be found in Marvig and colleagues [39].
(EPS)

**S1 Data. Dynamic exometabolomics data.** Values correspond to mM concentrations of the analyzed metabolites by their collection OD and strain.
(XLSX)

**S2 Data. Proteomic data for the clinical isolates.** Values correspond to the $Log_2$(Fold-change) for the indicated comparison. Statistical significance is reported by q values.
(XLSX)

**S3 Data. Proteomic data for the *aceE* and *aceF* recombinant strains.** Values correspond to the $Log_2$(Fold-change) for the indicated comparison. Statistical significance is reported by q values.
(XLSX)

**S4 Data. Individual numerical values underlying all figures.**
(XLSX)

**S1 Table. List of strains and plasmid.**
(DOCX)

**S2 Table. List of primers.**
(DOCX)

**S1 Text. Detailed information on metabolic preferences and convergent proteomic changes.**
(DOCX)

## Acknowledgments

The Basal Cell Immortalized Non-Smoker 1.1 (BCi-NS1.1) cell line was a kind gift from Professor Ronald G. Cristal (Weil Cornell Medical College, New York, USA).

## Author Contributions

**Conceptualization:** Søren Molin, Ruggero La Rosa.

**Data curation:** Bjarke Haldrup Pedersen, Ruggero La Rosa.

**Formal analysis:** Bjarke Haldrup Pedersen, Ruggero La Rosa.

**Funding acquisition:** Helle Krogh Johansen, Søren Molin.

**Investigation:** Bjarke Haldrup Pedersen, Filipa Bica Simões, Ivan Pogrebnyakov, Ruggero La Rosa.

**Methodology:** Martin Welch, Ruggero La Rosa.

**Project administration:** Søren Molin, Ruggero La Rosa.

**Resources:** Martin Welch, Helle Krogh Johansen, Søren Molin.

**Supervision:** Søren Molin, Ruggero La Rosa.

**Visualization:** Bjarke Haldrup Pedersen, Ruggero La Rosa.

**Writing – original draft:** Bjarke Haldrup Pedersen, Ruggero La Rosa.

**Writing – review & editing:** Martin Welch, Helle Krogh Johansen, Søren Molin, Ruggero La Rosa.

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
