## [Editor Report · Decision Letter 0]

23 Jan 2024

Dear Dr La Rosa, 

Thank you for submitting your manuscript entitled "Metabolic specialization drives reduced pathogenicity in Pseudomonas aeruginosa CF clinical isolates" for consideration as a Research Article by PLOS Biology. I am so sorry for taking so long to come back to you.

Your manuscript has now been evaluated by the PLOS Biology editorial staff, as well as by an academic editor with relevant expertise, and I am writing to let you know that we would like to send your submission out for external peer review.

Once your full submission is complete, your paper will undergo a series of checks in preparation for peer review. After your manuscript has passed the checks it will be sent out for review. To provide the metadata for your submission, please Login to Editorial Manager (https://www.editorialmanager.com/pbiology) within two working days, i.e. by Jan 25 2024 11:59PM.

Kind regards,

Melissa

Melissa Vazquez Hernandez, Ph.D.

Associate Editor

PLOS Biology

---

## [Decision Letter · Decision Letter 1]

14 Mar 2024

Dear Dr La Rosa,

Thank you for your patience while your manuscript "Metabolic specialization drives reduced pathogenicity in Pseudomonas aeruginosa CF clinical isolates" was peer-reviewed at PLOS Biology. It has now been evaluated by the PLOS Biology editors, an Academic Editor with relevant expertise, and by three independent reviewers. 

In light of the reviews, which you will find at the end of this email, we would like to invite you to revise the work to thoroughly address the reviewers' reports.

As you will see, the reviewers are generally positive about your study and think it is timely and well done, but they raise several overlapping concerns that should be addressed for further consideration at PLOS Biology. Specifically, Reviewer #1 raises concerns about the lack of information for the statistical methods used and suggests additional testable hypotheses to evaluate the T3SS activity in the mutants. In addition, Reviewer #2 notes that the findings should be further supported with genetic complementation of the mutants and asks that phylogenetic data be provided to understand how genetically related the isolated strains are. Finally, Reviewer #3 provides several suggestions to improve the overall quality of the presentation and clarity of the manuscript.

Given the extent of revision needed, we cannot make a decision about publication until we have seen the revised manuscript and your response to the reviewers' comments. Your revised manuscript is likely to be sent for further evaluation by all or a subset of the reviewers.

**IMPORTANT - SUBMITTING YOUR REVISION**

*Re-submission Checklist*

*Published Peer Review*

*PLOS Data Policy*

*Blot and Gel Data Policy*

Sincerely,

Melissa

Melissa Vazquez Hernandez, Ph.D.

Associate Editor

PLOS Biology

REVIEWERS' COMMENTS

Reviewer #1: 

Pedersen et al. present a compelling manuscript examining if metabolic adaptation contributes to the persistence of P. aeruginosa (Pa) in chronic infections, like those in people with CF. The authors use a combination of metabolomics and proteomics to examine this concept using a well-characterized longitudinal collection of clinical isolates of Pa from pwCF. The manuscript is well written and the hypothesis is a new contribution to the field, especially as considered using the metabolomics approach with the clinical isolate collection, but there are a few comments for the authors consideration that if addressed, would strengthen the results presented.

Comments:

1. Overall, it is unclear the statistical methods used, if any, for a number of the observations. For example, for the metabotypes, which statistical tests allowed the authors to define these? For figure 3d, e, these are great visualizations of complex data. Were statistics performed on the highlighted phenotypes with differently regulated metabolites? If so, which are significant? Showing patterns in pathways is interesting, but statistics are needed to validate observations. This goes for most figures, please describe the statistical tests used for each figure, as they are likely quite different, given the many approaches used in the study. Without this information, it is very hard to rigorously review the study.

2. Some terminology is confusing. The authors use the term "naïve" metabotypes to describe cluster A, yet this cluster has a combination of early and late isolates. The term "adapted" also is used for other metabotypes. These terms when applied without statistical tests shape the conclusions and need to be rigorous tested to be used. Especially as naïve vs adapted clusters are driving later experiments and greatly impacting the conclusions drawn from the study.

3. In figure 2b, what is driving the two clusters of late isolates? Can you deduce this from the variables driving PC2 in fig 2A?

4. In figure 3, are there any patient characteristics unique to cluster B? Antibiotic regimen differences? Mucoidy of the isolates? Lung function? 

5. In figure 4b, what does the fold change look like for the early vs late isolates? This comparison seems very interesting to the study and would be nice to see, in addition to the PAO1 comparison shown.

6. Lines 257-258, this statement seems like a testable hypothesis. This comment should be either be tested experimentally or moved to the discussion, if there is no data.

7. Also lines 356-358, this commentary is more appropriate for the discussion than in the results, as it doesn't set up data to be shown.

8. In general, the authors are using the term immunogenicity, when they are just measuring IL-8 secretion. Inflammation is more descriptive of the results presented, since the authors are just focusing on IL-8.

9. The final paragraph in the results is interesting, but needs more experimental data to support the conclusions the authors are making. Does an aceE/F deletion mutant have altered T3SS activity? Changes in exotoxin gene expression? Mutations in both the PDHc pathway and T3SS in clinical isolates can be true and not related. Experimentation is needed to link these two in the strains discussed, if the authors want to present the conclusions in lines 373-376. Or move this commentary to the discussion section.

10. What is the pyruvate concentration in CF sputum? Addition of this information would support the authors results for a role for pyruvate in establishing persistent Pa infections in CF.

11. For the comment in lines 450-452, were antibiotic resistance genes dysregulated in the aceE or aceF proteomics studies? If the metabolic phenotype precedes the development of antimicrobial resistance, you might expect antibiotic resistance gene expression to be altered in the aceE/F mutant backgrounds.

Reviewer #2: 

The manuscript by Pedersen et al tackles the role that metabolic adaptation plays in the cystic fibrosis context. The manuscript first uses a metabolomics approach to assess differences between early and late isolates from 8 patients. The metabolomic data suggests that late isolates often have similar metabolic profiles compared to their early counterparts. Proteomic data also supports similarity in late strains compared to early. Both metabolomic and proteomic analyses point to the secretion of pyruvate as a key factor associated with late strains. Analysis of the proteomic data suggests that late strains have predicted changes in virulence protein expression that agree with previous work. They also find mutations in genes that code for the pyruvate dehydrogenase complex (aceE and aceF). They then generate mutants in the PA01 background that harbor these mutations, which they then use for whole cell proteomics. The proteomic data suggest far reaching consequences for gene expression in these mutants, and they were defective for virulence as measured in an ALI model.

This is an interesting study that tries to tie together metabolic adaptation with other known adaptations, and is tantalizing in suggesting that metabolic changes likely precede other adaptations. The finding need to be supported with further clarifications, a clear exposition of limitations, and genetic complementation of the mutants. Overall the findings of this study, especially the metabolomics and proteomics could be presented much more clearly, and these sections could be shortened. 

Major concerns:

1. It is clear that heterogeneous populations of PA can coexist for long periods of time in pwCF. Therefore, the selection of single isolates from early and late is a limitation unless the investigators can say definitively that the late strain was not present at the early timepoint. Likewise, the assumption that early isolates are not adapted to the CF airway should also be treated as a limitation. 

2. The selection of early and late strains is a bit unclear. In the results, it seems as if the late strains were specifically chosen based on decreased growth rates. This might bias the data and confuse "late" for "growth retardation". 

3. The authors do not address the phylogenetic relatedness of the strain and only refer to previous work in ref 36. However, it would be helpful to present that data here in order to understand how closely each of the strains is related to each other to understand whether the observed changes in metabolism are generalizable across PA. It would also help with interpretation of HCA and the clustering of metabolomic and proteomic profiles. 

4. The aceF and aceE mutants should be complemented especially since many of the tests are on phenotypes not directly related to the action of these two enzymes. 

Minor concerns:

Line 81: change to "A few examples..."

Line 89: change to "if such a mechanism"

Line 90: Is undisclosed the right word?

Lines 134-135 It would also be important to know how many SNP differences were found between each pair of genomes.

Line 180: change to "strain."

Line 235-236: What does "directed towards mean" 

Line 272: The mutations are listed but not the haplotypes. Did each mutation occur in it's own allele or were some found together. Also, define "silent"

Lines 409-411. The contention that aceE and aceF mutants occur prior to T3SS mutations is not well supported by the data. This could be addressed using a phylogenetic approach and other existing whole genome sequences. 

Fig 5g is unclear. Why are the + and o at slightly different heights?

Reviewer #3:

The evolution of antibiotic resistance dominates the research landscape involving microbial pathogens, but the mechanisms of microbial persistence are similarly important. In this study, the authors examine mechanisms of persistence, with a focus on metabolic rewiring/specialisation. They demonstrate that metabolic rewiring widely occurs during adaptation to the host, and showcase an example that re-wiring metabolism related to pyruvate can enhance the persister phenotype in P. aeruginosa. To achieve this, the authors use a myriad of advanced techniques including metabolomics, proteomics, and confocal microscopy. 

Due to the small sample size used in the study (n = 8 for most groupings e.g. 'early vs late' and 'early/late vs PA01') many of the metabolomics and proteomics highlight correlations and trends, but are restricted to broad associations. However the results relating to Figs 5 and 6 - where the authors compare genomic sequence data for 34 patients, make targeted genomic alterations, and assess phenotypes in an environmental context that emulates the human lung - are focussed and highly interesting. In these results the authors show how mutations affecting pyruvate metabolism can directly change phenotypes associated with persistence, showcasing that metabolic rewiring mutations may not only occur to alleviate trade-offs from other mutations.

Overall I feel that the manuscript is timely and presents some exciting results. I have only a few comments and questions regarding the manuscript. I do not believe any of these comments are essential to complete prior to publication, but addressing these may help improve the clarity and readability:

1. Lines 90-93: "previous metabolic characterisations of clinical strains were carried out on only a limited number of isolates and/or on bacterial cultures on one specific growth phase". This statement requires citations.

2. Lines 153-155: "These metabotypes separate the late isolate from each respective early isolate and from the rest of the non-adapted metabotypes". I think using the word 'non-adapted' here is misleading. We do not know if they have not adapted, only that they haven't adapted with regards to the screened phenotypes. Please provide a statement clarifying this next to the 'non-adapted' description.

3. Lines 178-179: "To test hypothesis that the observed metabolic specialisation is partially rooted in changes in expression of proteins involved in cellular metabolism". Why would the hypothesis be that metabolism changes are only partially rooted in changes to protein expression? This statement seems to be written so that the hypothesis directly matches the results observed, rather than an organic hypothesis made at the beginning of the experiment. I would suggest either removing the word 'partially' or adding an explanation as to why we would expect proteins relating to metabolism to only partly govern metabolic changes.

4. Lines 179-180: Typo - remove the plural from "we analysed the proteome of each clinical strains".

5. Line 284: "Leads to a Phe->Ser amino acid change". Which amino acid position?

6. Lines 295-299: The authors describe here the lack of association between pathogenic phenotypes and the ace mutations, but they later describe how ace mutants exhibit different persister phenotypes. I was expecting an explanation for these discrepancies to feature more prominently in the discussion.

7. Fig 1: I would suggest swapping the order of Figs 1A and 1B. 1B contextualises 1A by showing the negative correlation between colonisation time in the lung and growth rate, which nicely outlines why we see a difference between the early and late growth rates shown in 1A. I don't see what it adds to the results by going after 1A, however.

8. Fig 2C: Why are there missing datapoints between some early and late conditions?

9. Fig 2D: Why are confidence intervals only shown for pyruvate late and formate late? If the confidence intervals fall within the thick lines in the other conditions - or if the other lines have no replicates - please describe this in the figure legend after confidence intervals are discussed.

10. Fig 3E: I would advocate moving this panel to the supplementary information, as I did not find this association particularly convincing. 740/2061 proteins exhibited differential expression (line 185), of which 235 belonged to those related to metabolism (line 196). When so many proteins exhibit differential expression, we should not be surprised that there are a handful of differences affecting this pathway between clinical strains and a laboratory strain. Similarly, we should also not be surprised to see differences between these strains and those that have spent several more years adapting in a host. How do we know that these changes are all adaptive, could some not be compensatory mutations or products of genetic drift? It may be that the pathways have been altered to "optimise" metabolism in the lung, but as currently presented I think the authors draw too many conclusions from this panel.

---

## [Editor Report · Decision Letter 2]

16 Jul 2024

Dear Dr La Rosa,

Thank you for your patience while we considered your revised manuscript "Metabolic specialization drives reduced pathogenicity in Pseudomonas aeruginosa CF clinical isolates" for publication as a Research Article at PLOS Biology. This revised version of your manuscript has been evaluated by the PLOS Biology editors and the Academic Editor.

Based on our Academic Editor's assessment of your revision, we are likely to accept this manuscript for publication, provided you satisfactorily address the remaining editorial points. Please also make sure to address the following data and other policy-related requests.

a) We routinely suggest changes to titles to ensure maximum accessibility for a broad, non-specialist readership, and to ensure they reflect the contents of the paper. In this case, we would suggest a minor edit to the title, as follows. Please ensure you change both the manuscript file and the online submission system, as they need to match for final acceptance:

"Metabolic specialization drives reduced pathogenicity in Pseudomonas aeruginosa isolates from cystic fibrosis patients"

Please supply the numerical values either in the a supplementary file or as a permanent DOI’d deposition for the following figures:

Figure 1ABC, 2ACDE, 3AD, 4ABC, 5BCDEG, 6AB, S1A, S2CD, S4AB, S6ABCDEF, S7BCD, S8.

c) Please cite the location of the data clearly in all relevant main and supplementary Figure legends, e.g. “The data underlying this Figure can be found in S1 Data” or “The data underlying this Figure can be found in https://doi.org/10.5281/zenodo.XXXXX”

d) We require the tree files for Figures 2B, 3BC, 4C, S4A

e) Please ensure that your Data Statement in the submission system accurately describes where your data can be found and is in final format, as it will be published as written there.

f) Per journal policy, if you have generated any custom code during the course of this investigation, please make it available without restrictions upon publication. Please ensure that the code is sufficiently well documented and reusable, and that your Data Statement in the Editorial Manager submission system accurately describes where your code can be found.

We expect to receive your revised manuscript within two weeks. 

*Published Peer Review History*

*Press*

Sincerely,

Melissa

Melissa Vazquez Hernandez, Ph.D.

Associate Editor

PLOS Biology

---

## [Editor Report · Decision Letter 3]

1 Aug 2024

Dear Ruggero,

Thank you for the submission of your revised Research Article "Metabolic specialization drives reduced pathogenicity in Pseudomonas aeruginosa isolates from cystic fibrosis" for publication in PLOS Biology. On behalf of my colleagues and the Academic Editor, Alice Prince, I am pleased to say that we can in principle accept your manuscript for publication, provided you address any remaining formatting and reporting issues. These will be detailed in an email you should receive within 2-3 business days from our colleagues in the journal operations team; no action is required from you until then. Please note that we will not be able to formally accept your manuscript and schedule it for publication until you have completed any requested changes.

PRESS

Sincerely, 

Melissa

Melissa Vazquez Hernandez, Ph.D., Ph.D.

Associate Editor

PLOS Biology
